# Proteomic sensors for quantitative multiplexed and spatial monitoring of kinase signaling

William J. Comstock[1,3], Marcos V. A. S. Navarro[1,2,3], Deanna V. Maybee[1], Yiseo Rho[1], Mateusz Wagner [1], Khoula Jaber[1], Yingzheng Wang[1] & Marcus B. Smolka [1]✉

Understanding kinase action requires precise quantitative measurements of their activity in vivo. In addition, the ability to capture spatial information of kinase activity is crucial to deconvolute complex signaling networks, interrogate multifaceted kinase actions, and assess drug effects or genetic perturbations. Here we develop a proteomic kinase activity sensor technique (ProKAS) for the analysis of kinase signaling using mass spectrometry. ProKAS is based on a tandem array of peptide sensors with amino acid barcodes that allow multiplexed analysis for spatial, kinetic, and screening applications. We engineered a ProKAS module to simultaneously monitor the activities of the DNA damage response kinases ATR, ATM, and CHK1 in response to genotoxic drugs, while also uncovering differences between these signaling responses in the nucleus, cytosol, and replication factories. Furthermore, we developed an in silico approach for the rational design of specific substrate peptides expandable to other kinases. Overall, ProKAS is a versatile system for systematically and spatially probing kinase action in cells.

Most cellular processes are controlled by the coordinated action of protein kinases. However, the multifaceted actions of over 500 kinases in human cells pose major challenges for the study of kinase function and the deconvolution of signaling networks in cells. A range of techniques are available for investigating in vivo kinase activity, although each technology suffers from important intrinsic limitations. Mass spectrometry-based phosphoproteomic technologies are capable of mapping and quantitatively analyzing tens of thousands of phosphorylation events in cells. However, without meticulous subcellular fractionation, they lack the ability to probe kinase action with spatial resolution, a key variable in the study of kinase biology[1,2]. Phosphoproteomic approaches are also not easily parallelizable, making them less amenable to high throughput analyses. Moreover, phosphoproteomic data do not inform on direct kinase-substrate relationships unless combined with computational motif analyses,

whose assignments are often ambiguous and biased towards well-characterized kinases[3]. Phosphorylation site-specific antibodies have been traditionally used to image the location of a phosphorylation event in cells using microscopy[4,5]. Unfortunately, antibodies displaying high specificity and low background for a given phosphorylation site are difficult to generate and not widely available. Therefore, experiments based on phospho-specific antibodies commonly suffer from low signal-to-noise ratio, limited dynamic range, and tend to yield semi-quantitative results that require extensive and complex image analyses and are not compatible with high-throughput analytical purposes.

Genetically encoded fluorescent biosensors offer a solution for the direct visualization of kinase activity toward specific target peptides in living cells[6,7]. Fluorescent kinase sensors, typically designed with a pair of fluorescent proteins for Förster Resonance

[1]Weill Institute for Cell and Molecular Biology, Department of Molecular Biology and Genetics, Cornell University, Ithaca, NY, USA. [2]IFSC Institute of Physics of São Carlos, University of São Paulo, São Carlos, São Paulo, Brazil. [3]These authors contributed equally: William J. Comstock, Marcos V. A. S. Navarro. ✉e-mail: mbs266@cornell.edu

Energy Transfer (FRET) or a conformationally changing circularly permuted fluorescent protein (cpFP), integrate a kinase-specific peptide substrate with a phospho-motif binding domain[7–9]. Upon phosphorylation by the kinase of interest, the recognition of the phospho-peptide by the binding domain induces a conformational change, triggering changes in energy transfer in FRET-based sensors or fluorescence in cpFP-based sensors[9]. Alternatively, the phosphorylation of the peptide substrate sequence can cause translocation or degradation of the fluorophore[7]. Cell-penetrating fluorescent (FLIM) probes for Abl and Src-family kinases have also been designed and implemented without the need for FRET[10]. Despite the successful application of fluorescent kinase sensors in a range of biological contexts, these systems have limitations, both in terms of general applicability as well as technical implementation. The specificity required for recognition by the phospho-motif domain limits the range of kinases that can be monitored, and the biosensor design process is labor-intensive, often requiring extensive optimization[11]. Moreover, excessive binding affinity between the phospho-motif and the interaction domain can lead to sensor saturation, restricting dynamic range and potentially impeding dephosphorylation by phosphatases. This was observed with a designed ATM sensor, which showed reduced efficiency in monitoring phosphorylation decay following kinase inhibition due to excessive binding strength[12]. Additionally, multiplexing these sensors is constrained by fluorescence signal overlap, limiting the ability to track multiple kinase activities concurrently[11].

Damage to DNA or stress during DNA replication trigger the activation of phosphatidylinositol 3′ kinase (PI3K)-related kinases (PIKKs) ATR, ATM and DNA-PKcs[13–15]. Once activated, PIKKs orchestrate elaborate signaling responses that regulate a range of cellular processes such as DNA repair, DNA replication, the cell cycle, transcription, and apoptosis. The downstream kinases CHK1 and CHK2 are activated by ATR and ATM, respectively, and mediate key aspects of the overall signaling response, including replication fork stability and cell cycle progression[16,17]. Despite extensive studies on PIKKs and downstream signaling responses and the mapping of the signaling network controlled by these kinases, our understanding of the spatial organization of kinase signaling within distinct subnuclear domains and cellular compartments remains elusive. There is currently a need for quantitative tools capable of rigorously and systematically monitoring locations and kinetics of DNA damage signaling in a lesion- and cell type-dependent manner, and with high dynamic range and throughput.

Here, we develop the Proteomic Kinase Activity Sensor (ProKAS) technique that leverages MS for the multiplexed, spatially resolved, and quantitative monitoring of kinase activity in living cells. ProKAS is based on a tandem array of peptide sensors that allows simultaneous tracking of multiple kinases within a single polypeptide module. The introduction of amino acid barcodes into these peptide substrates enables the multiplexed monitoring of kinase activities across different cellular locations or under varying experimental conditions. The multiplexing capabilities make ProKAS compatible with high-throughput analyses and screening purposes. We engineered a ProKAS module with sensors specifically designed to sense the activity of DNA damage response kinases ATR, ATM, and CHK1, with an expanded version also including a pan-CDK sensor and a sensor for the PPM1D phosphatase. Our results demonstrate the ability of the ProKAS sensor to capture kinase activity with high specificity and spatio-temporal resolution, uncovering kinetics of kinase signaling during DNA damage responses. Additionally, we developed a de novo approach for the rational design of substrate peptides, which is expected to be broadly applicable to most kinases within the human kinome. Overall, ProKAS offers a versatile method for probing entire kinase signaling networks, opening additional avenues for investigating kinase action in cells.

## Results

### ProKAS biosensor design

We designed the Proteomic Kinase Activity Sensor (ProKAS) technique to allow for quantitative, multiplexable and spatially resolved monitoring of kinase activity in cells using mass spectrometry (MS) (Fig. 1). The core of ProKAS lies in the Multiplexed Kinase Sensor (MKS) module, a tandem array of 10-15 amino acid peptide sensors representing preferred substrate motifs for selected kinases of interest (KOIs) (Fig. 1A, B). Each sensor has a serine or threonine at its center and three or more small amino acid residues at the edge for barcoding purposes (Fig. 1B). The ProKAS polypeptide also features an N-terminal enhanced Green Fluorescent Protein (eGFP) for imaging and a Targeting Element (TE) that directs the sensor to a specific subcellular location (e.g. nuclear localization signal (NLS), nuclear export signal (NES), or a protein of interest) (Supplementary Data 1). To enable proteomic analysis, we incorporated an affinity tag (e.g., ALFA tag) and also flanked each kinase substrate motif in the MKS with arginine residues, ensuring that trypsin digestion during sample preparation would result in kinase sensor peptides with distinct masses readily identifiable and quantifiable via MS[18]. The sensor peptides were also checked to ensure the target serine or threonine was not immediately flanked by lysines or arginines, which would result in tryptic peptides too small for conventional detection. The experimental workflow involves transfecting cells in culture with plasmids expressing the ProKAS biosensor with kinase sensor peptides and targeting elements of choice, after which cells are treated with a specific stimulus (Fig. 1C). Following cell lysis and affinity purification, tryptic digestion generates a mixture of modified and unmodified sensor peptides that are quantified via MS, assessing the level to which each kinase sensor peptide became phosphorylated, normalized by the abundance of the unmodified version of each sensor peptide. The use of barcodes allows the generation of ProKAS biosensor libraries and multiplexed analyses where each code can be linked to a specific targeting element, enabling analysis with spatial resolution (Fig. 1D). Alternatively, barcodes can be linked to other variables, such as distinct drug treatments or genotypes, enabling comprehensive profiling of kinase signaling networks and facilitating high-throughput screens for drug effects or genetic perturbations.

### Generation of ProKAS sensors based on endogenous kinase substrates

The success of ProKAS relies on high-quality sensors used in the MKS module, wherein each peptide substrate must report the activity of one kinase with sensitivity and specificity. We reasoned that the sequence of the substrate peptides could be based on -10-15 amino acid residues flanking phosphorylation sites on endogenous substrates, which could provide enough specificity for preferential targeting by a KOI. Phosphoproteomic data from large-scale experiments using kinase inhibitors or loss-of-function mutations would provide initial lists of phosphorylation events uniquely dependent on the KOI that are also compatible with MS detection. Once cloned into ProKAS biosensors, experiments should validate: (1) the ability of the MS to detect the phosphorylated peptide sensor; (2) the stimulus-dependent regulation of the probe phosphorylation; and (3) the specificity of the probe phosphorylation by the KOI (Fig. 2A).

As a proof of principle, we utilized the pipeline depicted in Fig. 2A to obtain MS-detectable ATR substrates suitable for use as ATR kinase sensor peptides (Fig. 2B). Large-scale phosphoproteomic analyses of camptothecin (CPT)-treated cells revealed dozens of phosphorylation sites in established ATR substrates that were strongly impaired by the ATR inhibitor, indicating high specificity for ATR and no predominant targeting by related kinases ATM or DNA-PKcs. Among these sites was serine 717 of FANCD2, a previously known ATR substrate, which our data indicated was highly dependent on ATR and readily detectable via

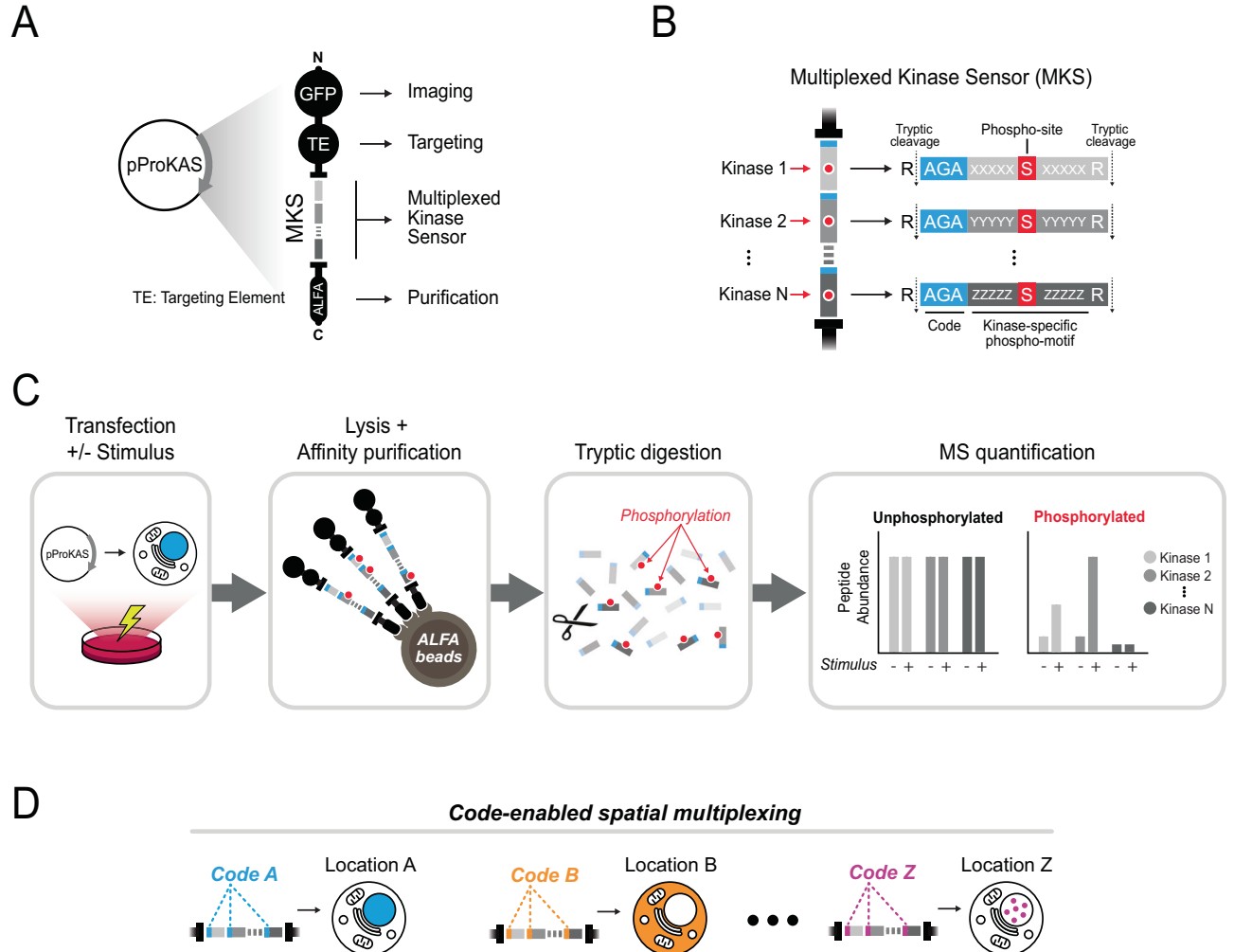

**Fig. 1 | Design and rationale of ProKAS, a modular technique for multiplexed analysis of kinase activity using mass spectrometry. A** Schematic representation of the ProKAS construct for expression of the biosensor polypeptide containing fluorescent tags for visualization, a targeting element (TE) for subcellular localization, a multiplexed kinase sensor (MKS) module for detecting kinase activity, and an ALFA tag for high-affinity purification. **B** Detailed view of the MKS module consisted of multiple kinase sensors, each featuring a kinase substrate motif, a barcode, and a flanking arginine residue allowing for tryptic digestion and independent MS-based quantification. **C** Overview of the ProKAS workflow, starting with expression of biosensors via plasmid transfection. Expressing cell populations are then treated with a kinase stimulus of choice, after which ProKAS biosensors are purified from cell lysates, digested with trypsin, and the individual kinase sensors are quantified via MS in both their unphosphorylated and phosphorylated forms. **D** Application of ProKAS for multiplexed spatial analysis of kinase activity. Multiple plasmids with distinct TEs matched to a specific barcode are co-transfected. MS analysis distinguishes the sensors based on the barcode mass, allowing matching signal intensity of the specific peptide-probe to the respective cellular location.

mass spectrometry[19] (Fig. 2C, Supplementary Data 3, 4). We therefore selected the 13 amino acid sequence surrounding FANCD2 S[717] and designed the ATR sensor by also adding a barcode and flanking R residues. This design results in the generation of a tryptic phospho-peptide slightly different from the endogenous FANCD2 counterpart. We then cloned this sequence into a ProKAS vector bearing an NLS as the targeting element, intending to monitor ATR activity in the nucleus (Fig. 2D). After transfection into HEK293T cells and treatment with CPT, the ATR ProKAS biosensor was pulled down with anti-ALFA beads, trypsinized, and analyzed by MS. PRM quantitation was used to compare peak areas for both the unphosphorylated and phosphorylated versions of the ATR sensor candidate in conditions treated with drug vehicle and hydroxyurea (HU). This revealed a significant increase in the abundance of phosphorylated ATR sensor upon genotoxin treatment after normalizing by the abundance of the unphosphorylated form of the sensor (Fig. 2E). Sensor specificity was also confirmed by treating cells with ATRi prior to the addition of HU, which prevented HU-induced phosphorylation of the sensor and further reduced its phosphorylation status to below that of the untreated cells, likely due to inhibition of basal ATR activity (Fig. 2F). Overall, these results validate the strategy of extracting small (10-15 amino acids) sequences from endogenous substrates to design a ProKAS sensor that exhibits high specificity for a given KOI.

## De novo generation of a kinase-specific sensor peptide for ProKAS

We also developed a computational approach for the design of specific kinase probes by leveraging a dataset derived from positional scanning peptide array (PSPA) analyses of 303 human Ser/Thr kinases[20]. The pipeline shown in Fig. 3 scans PSPA-based kinase preference scores in a space of billions of 10-residue sequences using a genetic algorithm approach to identify those exhibiting high specificity for a KOI, while minimizing cross-reactivity with the broader kinome (Fig. 3A). After selection of 10 sequences with highest predicted specificity and lowest cross-reactivity, the candidate sequences are then cloned into and expressed as one single ProKAS biosensor for experimental validation (Fig. 3B). This allows for the identification of the sensor sequences that are detectable in the MS and that demonstrate the best kinase

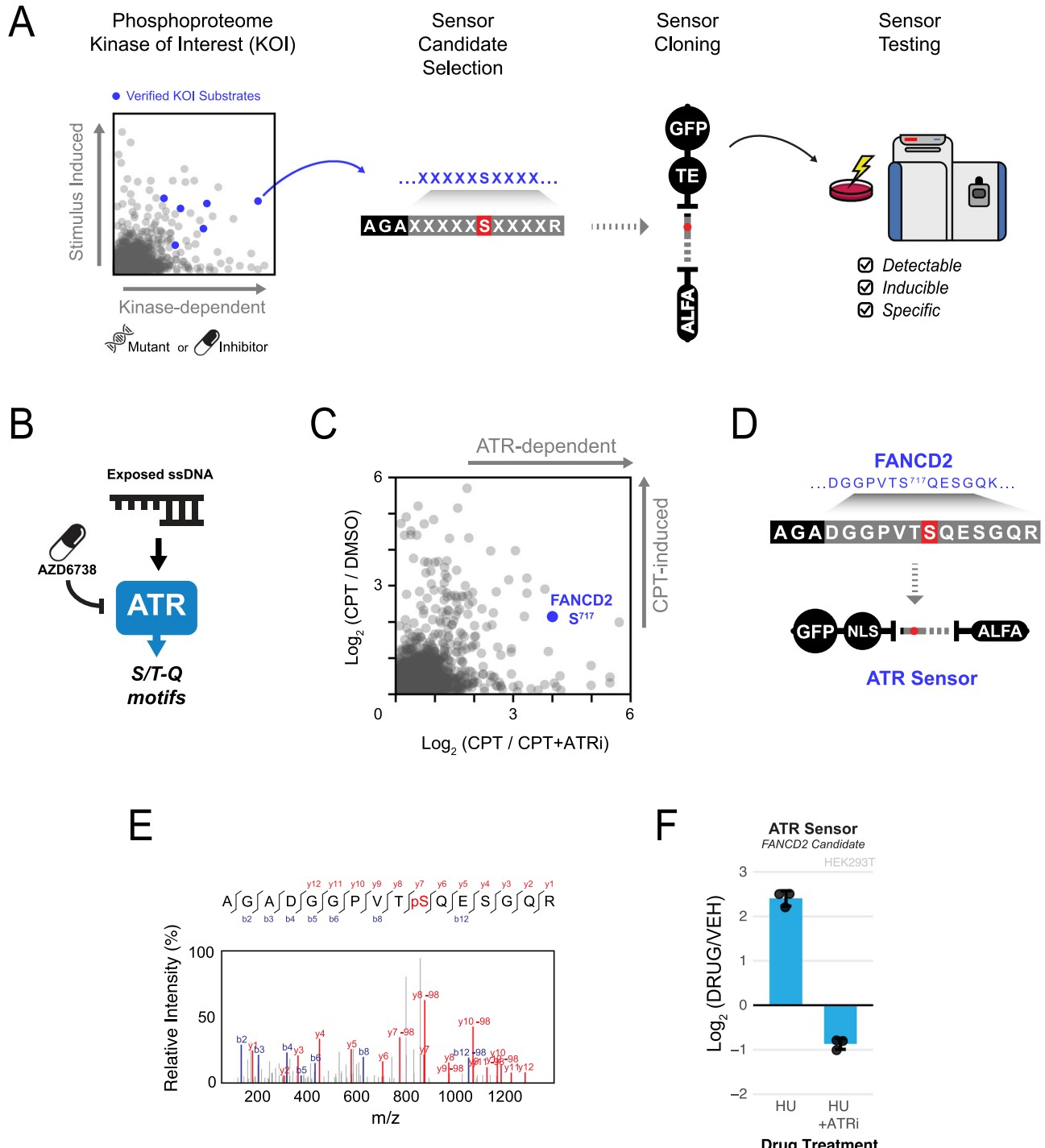

**Fig. 2 | Development and validation of a ProKAS sensor specific for ATR using phosphoproteomic data. A** Schematic outlining of the strategy for designing kinase sensors based on phosphoproteomic data and known endogenous substrates. 10-15 sequences surrounding phosphorylation events detected to be dependent on a kinase of interest (KOI) are cloned into ProKAS constructs, which are then expressed in cells to test for kinase sensor detectability, inducibility, and specificity. **B** Diagram showcasing that ATR is activated by single-stranded DNA damage, after which it preferentially phosphorylates substrates at the S/T-Q motif. Selective ATR inhibitors, such as AZD6738, are used in phosphoproteomic analyses to determine ATR-dependence of each detected phosphorylation event. **C** Identification of FANCD2 S717 as an ATR-specific phosphorylation site that is also

induced by genotoxin camptothecin (CPT) through phosphoproteomic analysis. Phosphoproteome results are included as Supplementary Data 3 and 4. **D** Cloning of sequence surrounding FANCD2 S717 as an ATR kinase sensor candidate into a ProKAS biosensor featuring a nuclear localization signal. **E** MS/MS spectrum gathered for phosphorylated ATR sensor, showcasing successful detection of the sensor candidate by MS. **F** MS analysis showing inducibility and specificity of the ATR sensor candidate after treatment with genotoxin and selective ATR inhibition, respectively. Cells were treated with 1 millimolar HU for 2 hours, and ATR-inhibited cells were treated with 5 micromolar AZD6738 15 minutes prior to HU addition. Error bars in F indicate the mean and standard deviation of triplicate independent experiments. Source data are provided as a Source Data file.

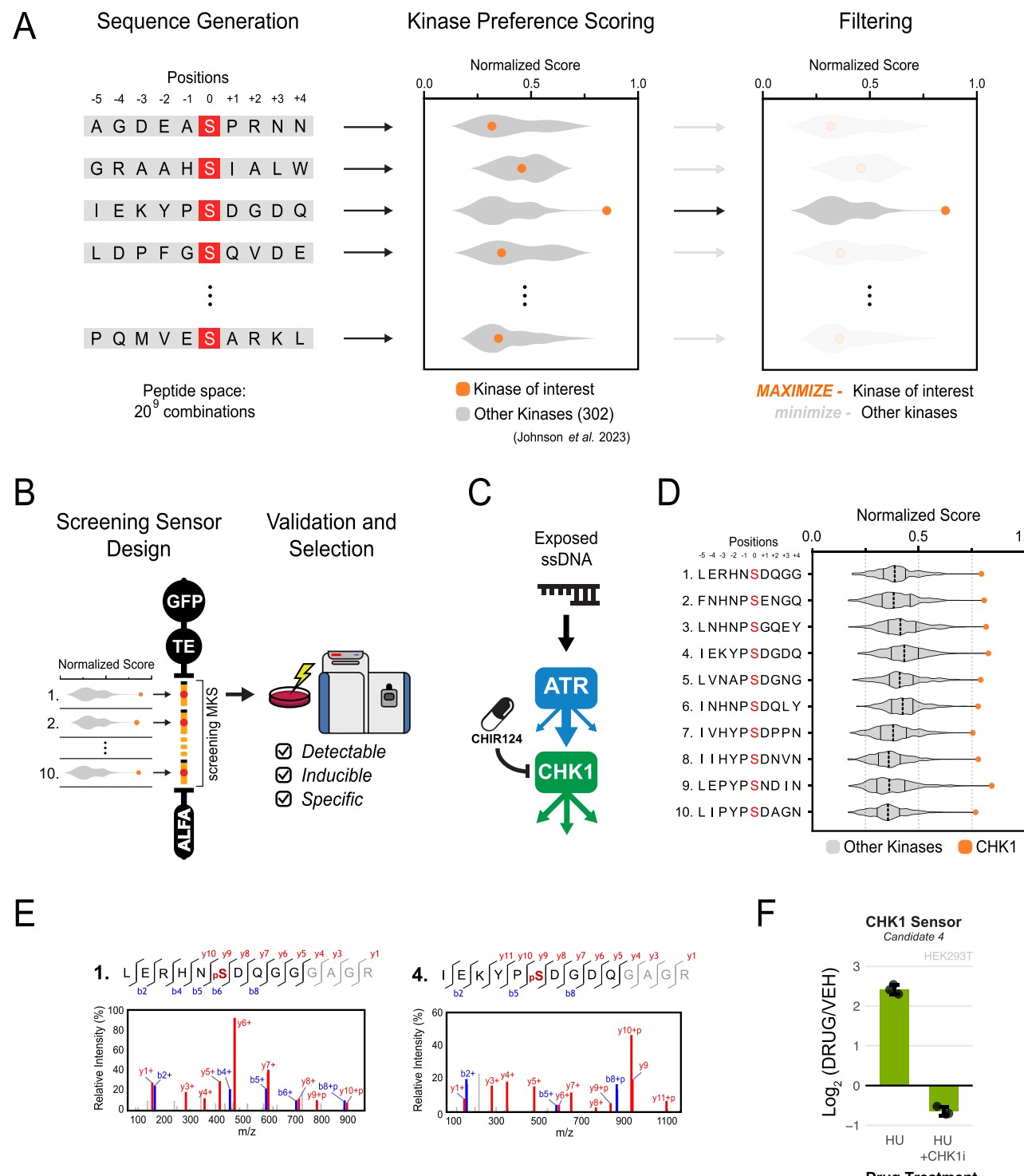

**Fig. 3 | Pipeline for the computational design and experimental validation of a CHK1-specific ProKAS sensor. A** Overview of the computational approach for generating kinase-specific motifs based on PSPA data. A genetic algorithm approach was utilized to arrive at a cohort of optimized sensor candidates out of the billions of possible sensor peptide sequences. **B** Workflow for multiplexed screening and selection of MS-detectable kinase sensors from candidates generated and filtered in silico. **C** Schematic illustrating CHK1 as a downstream effector kinase of ATR, and that CHK1 can be selectively inhibited by CHIR-124. **D** Selection of 10 candidate sequences for a CHK1 sensor based on PSPA scores as shown in (**A**).

**E** MS/MS spectra gathered for the phosphorylated form of the indicated CHK1 sensor candidates. Screening sensor with the candidates indicated in (**D**) were expressed in HEK293T cells treated with HU. **F** MS analysis showing inducibility and specificity of the CHK1-4 sensor candidate after treatment with genotoxin and selective CHK1 inhibition, respectively. Cells were treated with 1 millimolar HU for 2 hours, and CHK1-inhibited cells were treated with 500 nanomolar CHIR-124 15 minutes prior to HU addition. Error bars in G indicate the mean and standard deviation of triplicate independent experiments. Source data are provided as a Source Data file.

inducibility and specificity, ultimately guiding the selection of the top KOI-specific peptide substrate sensor to be used for ProKAS applications.

We applied this pipeline to develop a probe for CHK1, a key downstream effector kinase activated by ATR in response to DNA damage[21] (Fig. 3C). While CHK1 is known to preferentially phosphorylate substrates containing Arg or Lys residues at the −3 position relative to the phosphosite, the PSPA data confirmed this preference and further revealed that CHK1 exhibits strong preferences for bulky hydrophobic residues at the position −5 (e.g., Leu, Iso, and Phe), as well as a moderate preference for Pro and Asn at the position −1[20,22–24]. We selected sequences exhibiting high PSPA-bases scores for CHK1, while minimizing scores for the other 301 kinases (Fig. 3D). These contained Arg/Lys at position −3, but also incorporated variations at this and other positions to explore the impact on CHK1 specificity and sensor performance. A CHK1 ProKAS biosensor was generated with an MKS module containing 10 candidate sensor peptides and an NLS targeting element. MS analysis could readily detect tryptic peptides for all 10 candidate sensors (Supplementary Data 5). However, upon genotoxic stress, only two phosphorylated peptide substrates were identified, CHK1-1 (LERHNSDQGGGAGR) and CHK1-4 (IEKYPSDGDQGAGR), both containing Arg or Lys at the position −3 (Fig. 3E). The CHK1-4 peptide, selected as our final CHK1 sensor, demonstrated high inducibility upon cell treatment with HU, which is known to generate long stretches of ssDNA and thereby robustly activate ATR and consequently CHK1[13,25] (Fig. 3F). We also confirmed that the CHK1-4 sensor was specifically phosphorylated by CHK1 through the use of a selective CHK1 inhibitor CHIR-124[26] (Fig. 3F). Overall, these results confirm that Arg/Lys at the position −3 is a strong determinant of CHK1 specificity, and is consistent with the in vitro specificity assay showing that Leu/Iso at the position −5 favors CHK1 phosphorylation recognition in vivo. Importantly, in addition to generating preferential sequences for phosphorylation by the kinase of interest, this strategy is expected to also generate sequences that are less prone to be targeted by other kinases. The fact that phosphorylation of the sensor was impaired by CHK1 inhibitor supports the notion that the sensor is not being predominantly targeted by another kinase. These findings also highlight the need for an experimental screening step, given that the majority of in silico-generated sequences did not behave as effective sensors for our kinase of interest, whether due to a lack of phosphorylation or poor ionization efficiency of phosphopeptides. To compare with phosphoproteome-derived sensors, this approach was also applied to ATM, resulting in kinase sensor peptide sequences that proved to be MS-detectable, inducible, and specific to a similar degree when compared to the ATM sensor derived from our phosphoproteomic data (Supplementary Fig. 1).

## Multiplexed monitoring of ATR, ATM, and CHK1 activity with ProKAS

We next assessed the use of ProKAS for monitoring the activity of multiple kinases simultaneously, circumventing the need for conducting separate experiments for each kinase of interest. We generated a multiplexed "DDR" ProKAS biosensor to simultaneously monitor the activity of ATR, CHK1, and ATM kinases (Fig. 4A, B). For the ATM ProKAS sensor, we employed the phosphoproteome-guided strategy, revealing a phosphorylation event in the known ATM substrate 53BP1, which was ultimately selected as our sensor peptide[27] (Supplementary Fig. 2). The DDR ProKAS containing a tandem array of the ATR, ATM, and CHK1 sensors (Fig. 4B) was expressed in cells that were mock-treated or treated with either CPT or HU to predominantly generate DSBs or stalled forks, respectively[28,29]. We first confirmed that swapping substrate peptide positions within the MKS module did not alter their level of inducibility upon genotoxin-induced kinase activation (Supplementary Fig. 2). We next assessed

whether each of the sensors retained the specificity for the respective KOI. As shown in Fig. 4C, the use of specific kinase inhibitors confirmed that the sensors within the context of the ProKAS peptide array were specifically phosphorylated by the expected kinase. Inhibition of ATR under both genotoxin conditions revealed a reduction in the phosphorylation of the CHK1 sensor, consistent with the activation of CHK1 being canonically controlled by ATR[13,30]. Inhibition of CHK1 alone did not reduce the phosphorylation of ATR or ATM kinase sensors in cells treated with CPT. However, phosphorylation of the ATM sensor increased in HU-treated cells upon CHK1 inhibition, consistent with CHK1 playing key roles in preventing the collapse of stalled replication forks in HU, which causes DSBs and ATM activation[31]. As expected, ATM inhibition robustly and specifically ablated phosphorylation of the ATM sensor upon CPT treatment, and also displayed minor inhibition of the ATR and CHK1 sensors likely due to effects in inhibiting DNA end resection during the DSB response[32,33]. We also confirmed that expression of these kinase sensors did not impair the endogenous DNA damage response by blotting for markers of ATR and ATM activation after treatment with CPT and HU[34,35] (Supplementary Fig. 4). Expression of kinase sensors also had no significant impact on cell viability when measured via MTS assay (Supplementary Fig. 5). Additionally, large changes in expression level of the biosensors did not have a major effect on the quantitative readouts obtained from the kinase sensor peptides (Supplementary Fig. 6).

In addition to multiplexing, other advantages of ProKAS include the throughput and highly quantitative nature of the analyses, which should allow for precise kinetic studies on kinase signaling. We implemented a semi-automated pipeline for processing dozens of samples in 24-well cell culture plates (Fig. 4D), enabling temporal kinetic analysis of ATR, ATM, and CHK1 signaling in response to CPT and HU (Fig. 4E, F). Consistent with the expected behavior of these kinases upon treatment with CPT, which induces DSBs, ATM showed rapid and robust activation within 10 minutes, reflecting its role as a primary responder to DSBs. ATR activation followed, ultimately matching ATM activation levels at later timepoints, consistent with the longer time required for DNA ends to be resected and to support ssDNA-mediated ATR activation[36]. The slower kinetics of CHK1 activation were consistent with it being a kinase downstream of ATR. In contrast, HU treatment revealed distinct activation kinetics for these DDR kinases. Consistent with HU causing stalled replication forks and rapid ssDNA exposure, ATR was rapidly activated whereas the ATM activation did not reach levels as high as ATR activation (Fig. 4F). CHK1 phosphorylation was slower initially, but ultimately surpassed ATM. These observations align with the canonical behaviors of these kinases in response to DNA damage and highlight the ability of ProKAS to generate highly quantitative data on the kinetics of kinase activation. Notably, raw peak areas showed no significant drop in unmodified sensor abundance as treatment durations increased, suggesting that phosphorylation of the kinase sensor peptides does not lead to proteolytic degradation of the biosensor (Supplementary Figs. 7, 8 and Source Data). These results also showed that unmodified sensor peptides were several orders of magnitude more abundant than their phosphorylated counterparts, consistent with no perceptible depletion of the unmodified sensors as phosphorylation changes remain sub-stoichiometric[2]. Finally, we leveraged the multiplexed and throughput capabilities of ProKAS to assess the inhibitory potency of an ATM-specific inhibitor (AZD0156) in vivo. We performed an inhibitor titration experiment in HEK293T cells expressing the triplexed DDR ProKAS sensor, monitoring the phosphorylation levels of all three sensors simultaneously. As expected, AZD0156 potently inhibited ATM activity with an IC50 of 9 nanomolar, somewhat higher than what has been measured in HT29 cells via ATM autophophorylation[37] (Fig. 4G). Importantly, the inhibitor exhibited high selectivity, showing no significant

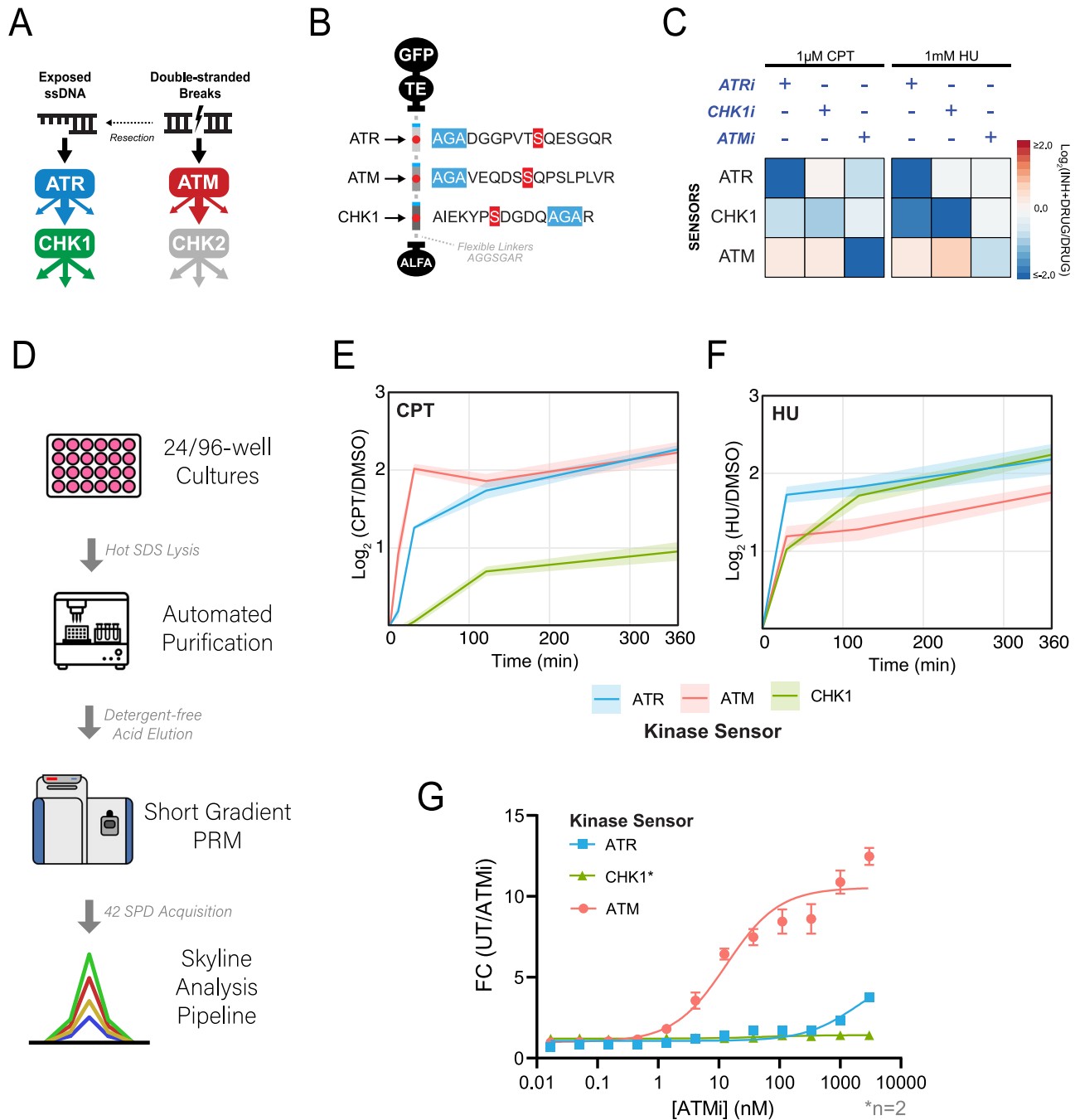

**Fig. 4 | Multiplexed analysis of DDR kinase activities using ProKAS. A** Schematic illustrating the canonical mechanisms of activation for DDR kinases ATR, ATM, and CHK1. **B** Design of a triplexed ProKAS construct containing kinase sensors for ATR, ATM, and CHK1. **C** Validation of kinase specificity of each sensor within the triplexed ProKAS construct via treatment with selective kinase inhibitors for each KOI in the presence of either 1 micromolar CPT or 1 millimolar HU. ATR, ATM, and CHK1 were inhibited with 5 micromolar AZD6738, 50 nanomolar AZD0156, and 500 nanomolar CHIR-124, respectively. **D** Flowchart showing primary components of the semi-automated high-throughput pipeline enabling larger-scale ProKAS experiments. **E** MS analysis showing kinase activation dynamics in response to 1 micromolar CPT over 6 hours in HEK293T cells expressing the triplexed ProKAS construct. **F** MS analysis showing kinase activation dynamics in response to 1 millimolar HU over 6 hours in HEK293T cells expressing the triplexed ProKAS construct. **G** MS analysis showing inhibitor titration using ProKAS to demonstrate the potency and selectivity of ATM inhibitor AZD0156. Cells were treated with 1 micromolar CPT for 30 minutes to activate ATM. Error bars/envelopes in **E**, **F** and **G** indicate the mean and standard deviation of triplicate independent experiments, except for CHK1 sensor quantification in G which comprises duplicate independent experiments from which no statistical significance has been calculated or displayed. Source data are provided as a Source Data file.

effect on CHK1 activity and minimal impact on ATR activity even at high concentrations (IC50 > 5 micromolar), aligning with the known degree of selectivity for this inhibitor. This experiment showcases the power of ProKAS in enabling precise, multiplexed, and in vivo assessment of kinase inhibitor efficacy and selectivity, offering a valuable tool for drug discovery and development.

## Spatial analysis of kinase signaling using location-barcoded ProKAS

In addition to its quantitative and multiplexed capabilities, ProKAS was designed to also enable the spatial analysis of kinase activity in cells. As shown in Fig. 1D, the ProKAS construct features a targeting element that controls cellular localization and can be linked to a specific amino

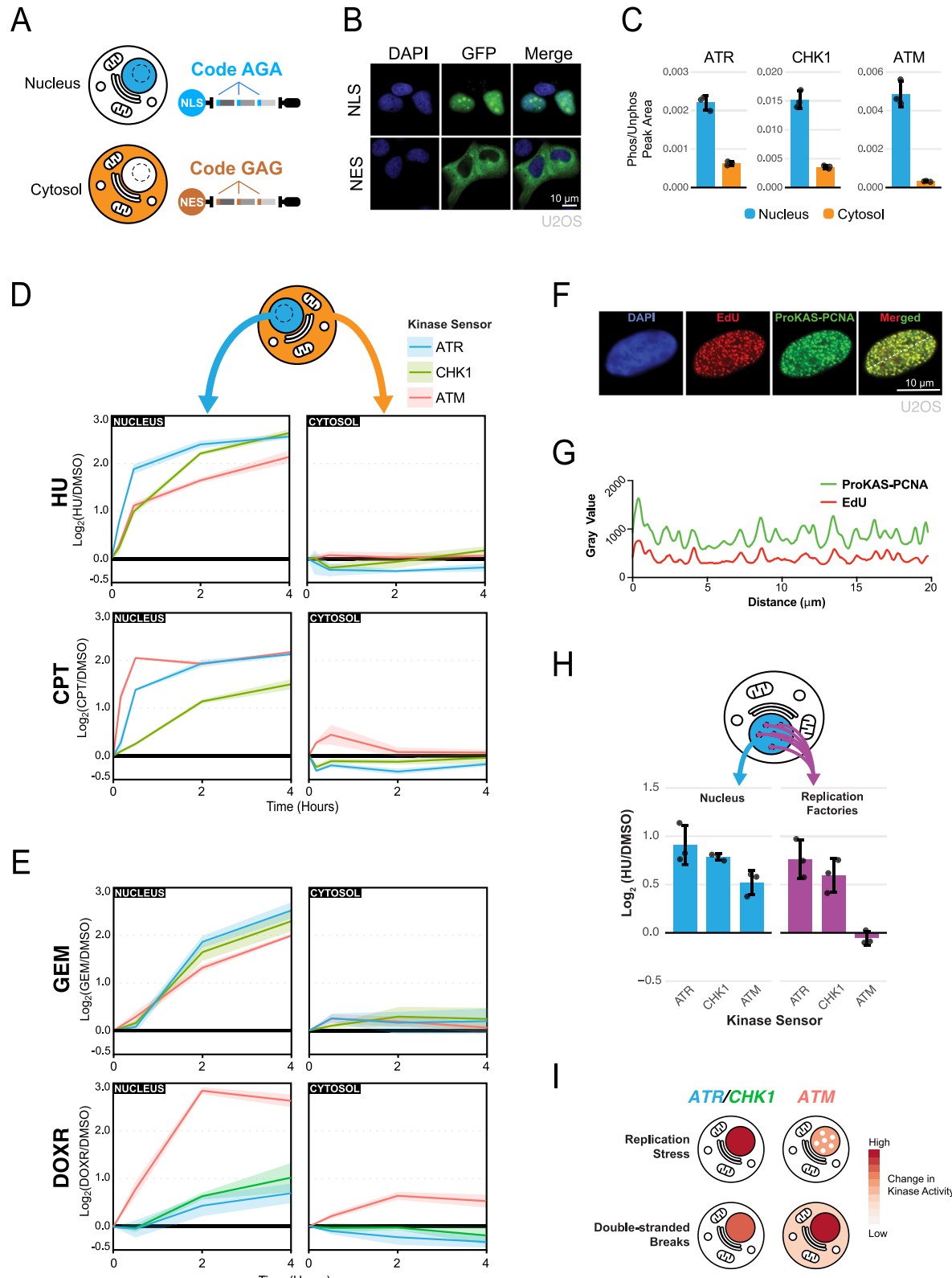

acid barcode embedded within each sensor peptide. Upon co-transfection of cells with ProKAS constructs containing different targeting elements, detection of distinct barcoded peptides by MS should reveal the relative levels of kinase activity in different cellular locations. As a proof of concept, we used an NLS or NES as targeting elements to direct the ProKAS polypeptide to the nucleus or cytosol, respectively (Fig. 5A, B). The AGA barcode was used in the construct containing the

NLS and a GAG barcode was used in the construct with the NES. Importantly, addition of different barcodes to the substrate sensor peptides within the MKS module did not alter their efficiency of phosphorylation upon kinase activation (Supplementary Fig. 9).

ATR, ATM, and CHK1 are DNA damage signaling kinases known to operate primarily inside the nucleus. While cytosolic roles for these kinases have been proposed[38-40], potential non-nuclear functions

**Fig. 5 | Spatially-encoded ProKAS for analysis of DDR kinase activities.**
**A** Illustration of nuclear and cytosolic ProKAS biosensors featuring distinct codes for simultaneous monitoring of DDR kinase activities in both locations.
**B** Microscopy showing the localization of ProKAS biosensors containing an NLS or an NES. Experiment was repeated 3 times. **C** Peak area ratios for DDR kinase sensors showing the proportion of phosphorylated sensor peak area to unphosphorylated peak area. Data based on MS analysis of cells not treated with any genotoxin. **D** MS analysis of co-expressed NLS- and NES-containing ProKAS biosensors, simultaneously monitoring DDR kinase signaling kinetics in both the nucleus and cytosol in response to 1 millimolar HU and 1 micromolar CPT. The experiments were quantified via SILAC. **E** Nuclear and cytosolic DDR kinase signaling kinetics were also monitored after treating cells with 10 micromolar gemcitabine (GEM) and 5

micromolar doxorubicin (DOXR). The experiments were quantified via label-free quantification (LFQ). **F** Microscopy showing the co-localization of the ProKAS biosensor (utilizing PCNA as a targeting element) with EdU foci. The experiment was repeated 3 times. **G** Densitometry analysis of EdU and ProKAS-PCNA signal across the white line drawn across the nucleus in panel **E**, showing signal coincidence between the EdU and ProKAS-PCNA foci formed. **H** MS analysis simultaneously monitoring the effect of HU on ProKAS biosensors containing an NLS or PCNA as the targeting element. **I** Schematic illustration of different spatial distributions of ATR, CHK1, and ATM kinase activity observed upon replication stress or DSBs. Error bars/envelopes in **C**, **D**, and **F** indicate the mean and standard deviation of triplicate independent experiments. Source data are provided as a Source Data file.

remain elusive due, in part, to the need of more quantitative tools capable of monitoring the activity of these kinases with rigorous specificity and spatial resolution. Using ProKAS, we analyzed cells co-transfected and thereby co-expressing nuclear and cytosolic versions of the multiplexed ATR, ATM, and CHK1 sensors differentially barcoded based on their location (Fig. 5C–H). The intensity of the phosphorylated peptide in each compartment was divided by the intensity of the corresponding unphosphorylated peptide in that compartment, yielding normalized values of peptide phosphorylation, a rough measure of stoichiometry of phosphorylation. As shown in Fig. 5C, while we were able to detect phosphorylation of all three sensors in both nucleus and cytosol in the absence of any genotoxic treatment, the constitutive levels of kinase sensor phosphorylation were significantly lower for the cytosol-localized peptide sensors compared to their nucleus-localized counterparts. Such high levels of constitutive kinase signaling in the nucleus are consistent with the high endogenous levels of DNA replication stress in HEK293T cells[41]. We next monitored the kinetics of ATR, ATM, and CHK1 signaling in the nucleus and in the cytosol following treatment with CPT and HU (Fig. 5D). Once again, the two drugs exhibited distinct patterns of kinase activation over time in the nucleus. Interestingly, while HU did not result in major changes in kinase activity in the cytosol, ATM activity was specifically detected in the cytosol when treating cells with CPT. We proceeded to perform similar kinetics experiments utilizing two more genotoxic agents, gemcitabine (GEM) and doxorubicin (DOXR), which again revealed distinct kinase activation dynamics (Fig. 5E). Despite being an inhibitor of ribonucleotide reductase like HU, GEM showed a slower pattern of kinase activation in the nucleus while still exhibiting a similar lack of cytosolic kinase activity changes. DOXR also showed different dynamics from CPT, even though both genotoxic agents are DSB-inducing toposiomerase inhibitors, inducing a strong increase in nuclear ATM signaling that was largely uncoupled from ATR-CHK1 signaling. However, as seen with CPT, an increase in cytosolic ATM signaling was once again detected, which was then shown to be prevented via ATM inhibition in the presence of both drugs (Supplementary Fig. 10). This cytosolic ATM sensor phosphorylation was verified via cytosolic fractionation followed by ALFA purification, which also confirmed that the ATR and CHK1 sensors featured no induced phosphorylation in the cytosol (Supplementary Fig. 11). ATM has been shown to localize to various cytosolic locations in specific situations[42,43], but its level of activation outside the nucleus has been difficult to measure. While the increase in cytosolic ATM signaling was relatively mild compared to the detected increase in nuclear ATM signaling, it is important to note that the larger volume of the cytosol is likely diluting the observed induction, especially if the source of the non-nuclear ATM activation is at a specific cellular location, such as the external membrane of a specific organelle. Since we did not detect an increase in cytosolic ATR or CHK1 signaling in response to any of the treatments tested, the increase in cytosolic ATM signaling induced by DSB is likely a specific feature of ATM activation or signaling propagation, and is unlikely to represent a non-selective leak of nuclear kinases into the cytosol.

To demonstrate the ability of ProKAS to probe differences in kinase activity within nuclear sub-compartments, we focused on sites of DNA synthesis. Upon HU treatment, sites of DNA synthesis are expected to accumulate stalled replication forks and single-stranded DNA, resulting in the selective induction of ATR-CHK1 signaling[15]. We localized the ProKAS biosensor to sites of DNA synthesis by using the processivity factor PCNA clamp, a key component of replication forks, as the targeting element[44]. Microscopy analysis confirmed that the PCNA-containing biosensors co-localized with nascent DNA, as visualized via incorporation of EdU (Fig. 5F, E). Using HCT116 cells, we then co-expressed ProKAS biosensors using PCNA with an NLS or only the NLS as the targeting element, allowing comparison of kinase signaling diffused in the nucleoplasm with signaling localized to sites of DNA synthesis. Upon treatment with 1 millimolar HU for 1 hour, biosensors localized to the nucleus with an NLS alone showed induced phosphorylation of all three kinase sensors (Fig. 5H). In contrast, the ATM sensor in the PCNA-containing ProKAS construct exhibited no induced phosphorylation upon HU treatment, while the ATR and CHK1 sensors were robustly induced. This finding is consistent with the notion that fork stalling specifically activates ATR-CHK1, but not ATM, which mainly responds to DSBs[13,14]. This finding also shows that while HU does result in ATM activation, the nature and location of such activation is different than where ATR-CHK1 are activated, and is unlikely to occur at stalled forks containing PCNA. Overall, these results demonstrate the ability of ProKAS to monitor kinase signaling with spatial resolution and define key differences in the cellular location of ATR and ATM signaling upon replication stress and DSBs (Fig. 5I).

**Expanded multiplexing capabilities of ProKAS**
We further expanded on the multiplexing power of ProKAS by extending the usable cohort of amino acid barcodes (Fig. 6A). As mentioned above, amino acid barcodes in the MKS linked to distinct targeting elements enabled the spatially-resolved analysis of several kinases simultaneously (Fig. 5), but we reasoned that amino acid barcodes could find applications outside of spatial multiplexing as well. We assessed a larger cohort of amino acid codes, ensuring that selected codes had negligible impact on kinase preferences (Fig. 6A, Supplementary Fig. 9). Capitalizing on other multiplexing capabilities of ProKAS, we also added more sensor peptides to the construct (Fig. 6A). We added a sensor to monitor the action of CDK1 and CDK2, designed by retrieving a known CDK substrate, SAMHD1 Threonine 592, from the phosphorylation site database PhosphositePlus[45–47]. Phosphorylation of the CDK sensor was reduced upon CDK inhibition and, consistent with the known behaviors of CDK1 and CDK2 during genotoxic responses, decreased after replication fork stalling (Supplementary Fig. 12). Additionally, ATR inhibition was found to increase the phosphorylation of this sensor, in agreement with the known roles of ATR in regulating CDK activity[48,49] (Supplementary Fig. 12). We also added a sensor peptide designed to monitor co-regulation by the ATM kinase and the PPM1D phosphatase, derived from their common substrate RAP80 serine 101 (Fig. 6A)[50].

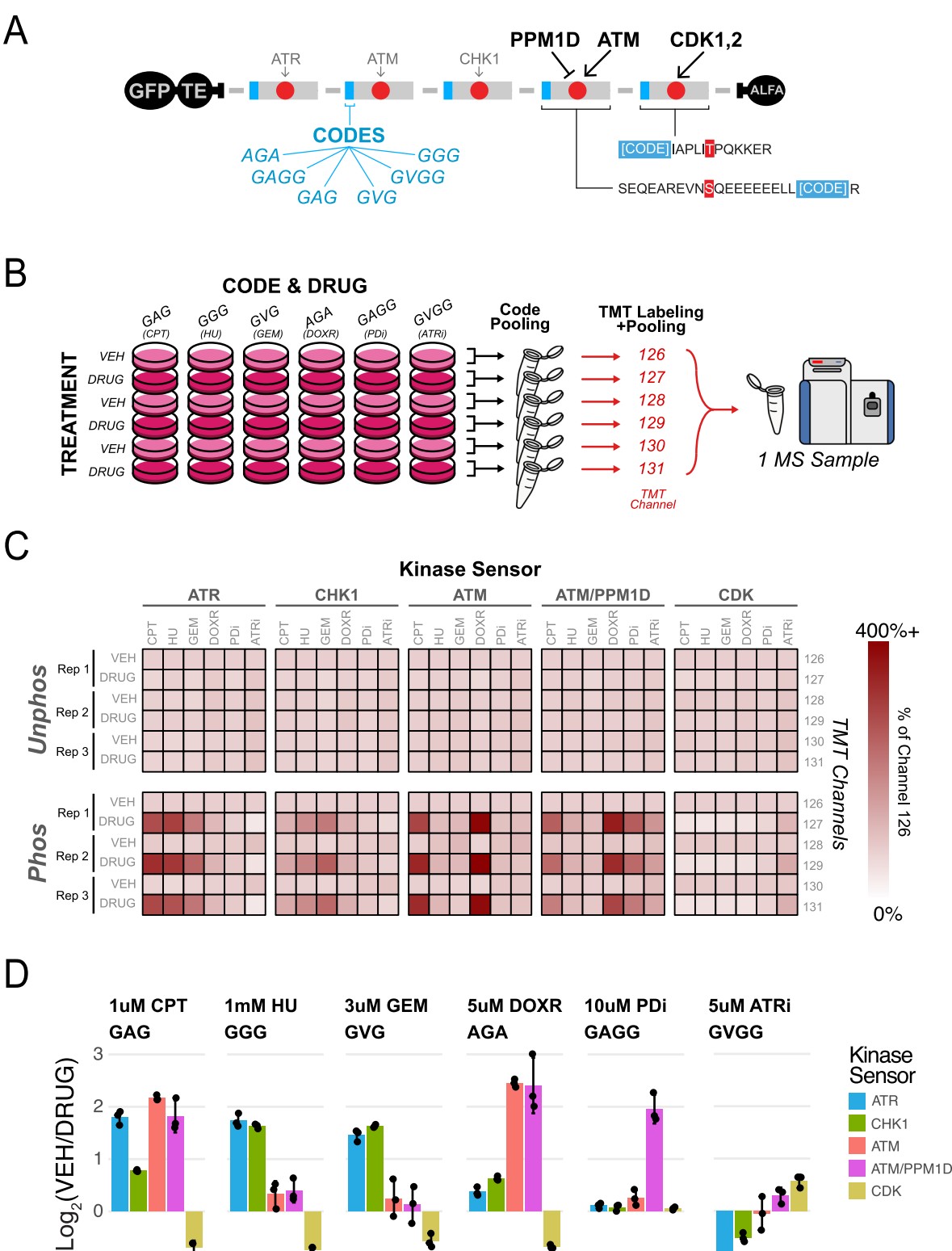

In addition to expanding multiplexing by increasing the number of sensors and barcodes, ProKAS-based mass spectrometry allows further multiplexing through the use of tandem mass tags (TMT)[51]. Since the barcodes confer a unique mass on each coded kinase sensor peptide, the number of conditions that could be monitored after incorporating TMT into the experimental pipeline would be equal to the number of codes multiplied by the number of TMT channels. To

demonstrate these expansions in multiplexing capabilities, we utilized the six validated barcodes in conjunction with TMT 6-plex to achieve hyperplexed quantification of 36 different experimental conditions, monitoring the behavior of 5 sensor peptides in each one (Fig. 6B). We grew 36 separate cultures of HEK293T cells, each receiving specified drug or vehicle treatment. With six cultures (one for each TMT channel) for each of the six codes, three were treated with a drug and three

**Fig. 6 | Expanded multiplexing capabilities via additional codes, sensor and TMT quantification. A** Schematic illustrating an expanded ProKAS biosensor featuring added sensors: one targeted by the ATM kinase and PPM1D phosphatase, and another targeted by CDK1 and 2. A larger cohort of amino acid codes is also depicted, expanded from three to six total codes. **B** Experimental design combining the expanded cohort of codes with TMT 6plex, allowing for the multiplexed analysis of 36 experimental conditions in one MS sample. Separate cell cultures were treated with drug or vehicle in triplicate, after which lysates with different codes were pooled to perform 6 affinity purifications. Tryptic digests from these purifications were then labeled with different TMT reagents before being pooled into a

single sample for analysis via LC-MS/MS. Drug treatments were 2 hours in duration at the following doses: 1 micromolar camptothecin (CPT), 1 millimolar hydroxyurea (HU), 3 micromolar gemcitabine (GEM), 5 micromolar doxorubicin (DOXR), 10 micromolar PPM1D inhibitor (PDi), and 5 micromolar ATR inhibitor (ATRi). **C** Matrices displaying TMT reporter channel relative abundances for the unphosphorylated and phosphorylated forms of the sensor peptides. **D** TMT-based MS quantification for each sensor peptide in each code, with each code corresponding to a given drug treatment. Error bars in D indicate the mean and standard deviation of triplicate independent experiments. Source data are provided as a Source Data file.

with vehicle for a duration of 2 hours, enabling biological triplicates to be quantified all within one MS sample (Fig. 6B). Cell lysates with different codes were pooled to reduce the number of affinity purifications performed (6 instead of 36), after which the eluted proteins were digested with trypsin and labeled with one of six TMT reagents, ultimately pooling into a single sample. Quantitative MS analysis revealed that unphosphorylated peptides featured uniform TMT quantification across all six channels, while the phosphorylated peptides were differentially regulated depending on the drug treatment (Fig. 6C). Fold changes were calculated for both forms of each sensor peptide across all six drug treatments performed, with the ratios obtained for the phosphorylated forms being normalized by that of the unphosphorylated forms (Fig. 6D). Consistent with aforementioned experiments, HU and GEM resulted in primarily ATR and CHK1 kinase activation, whereas CPT and DOXR again caused primarily ATM activity. However, the expanded cohort of sensor peptides also revealed that while in most cases the PPM1D sensor behaved similarly to the existing ATM sensor, PPM1D inhibition (PDi) via treatment with GSK2830371 resulted in a selective increase in ATM/PPM1D sensor phosphorylation, consistent with previous findings about the preferred motif of PPM1D[52,53]. Additionally, ATR inhibition was found to decrease levels of both ATR and CHK1 sensor phosphorylation while simultaneously increasing CDK sensor phosphorylation, once again consistent with the known roles of ATR signaling in CDK regulation[48,54]. This dataset, acquired from one MS sample, demonstrates that expanding a ProKAS biosensor in both the number of amino acid barcodes and sensor peptides, while also utilizing isobaric labeling like TMT, can enable highly multiplexed analyses, significantly augmenting the amount of information gathered from one ProKAS experiment and vastly increasing throughput. Conceptually, this logic could be further expanded to dozens of sensors and dozens (or even hundreds) of barcodes, allowing the concomitant analysis of hundreds of samples using ProKAS.

## Discussion

In the last decades, technological advances such as genetically encoded fluorescent biosensors, analog-sensitive engineered kinases, and MS-based phosphoproteomics have revolutionized our ability to study kinase action in cells[1,6,9,40,55]. However, much is yet unknown about how kinases act in cells to regulate and coordinate cellular processes and responses. A major obstacle is the lack of versatile and generalizable techniques capable of systematically integrating quantitative, spatial, and temporal analyses of kinase action.

The ProKAS system introduced here expands the set of tools available for monitoring kinase activity in living cells. ProKAS overcomes some of the limitations associated with traditional fluorescence-based kinase sensors by integrating the strengths of MS with the versatility of peptide-based biosensors. Many fluorescence-based kinase sensors involve elaborate engineering that is constrained not only by the identification of substrate peptide sequences that are specific to the KOI, but also by the requirement that, upon phosphorylation, the sequences are recognized by a phosphorylation-binding domain (e.x., FHA domain for FRET sensors) or induce changes in subcellular localization (e.x., nucleus to cytosolic translocation)[7]. A

major advantage of ProKAS is the flexibility in the generation of the sensor sequences, which are constrained mainly by the KOI specificity and by the ability to be detected by MS. Since sensors can be designed based on phospho-peptides detected in phosphoproteomic datasets, the requirement of MS-detectability is already satisfied upon phosphopeptide selection, whether from datasets generated in-house or retrieved from existing proteomic data repositories. In the case of sensors designed in silico, our experience is that most sequences of around 10 amino acids can generally be detected by MS, allowing flexibility in sequence design, though selection then becomes contingent on observed kinase specificity. We expect that the reduced stringencies in sequence design compared to FRET sensor design should make ProKAS a more generalizable tool for exploring the action of the broader kinome. We also observed higher dynamic ranges in the recorded ProKAS sensors compared to FRET sensors. For example, the recently reported FRET sensor for CHK1 showed a lower than 2-fold change in signal upon replication stress (typical of FRET sensors), whereas our CHK1 sensor displayed over 4-fold increase in phosphorylation upon replication stress[56]. It is also worth noting that ProKAS experiments were able to be conducted at small scale relative to phophoproteomic screens (~500 K cells versus ~200 million) and without performing gene editing to alter localization of any endogenous kinase substrates, which are functional proteins that could disrupt cell biology if tampered with.

Another important feature of ProKAS is the ability to monitor multiple kinases simultaneously, which is expected to be essential for studying complex signaling responses and for high-throughput analyses. In addition to having multiple kinase sensor peptides in one ProKAS biosensor, the use of amino acid barcodes further expands the possibilities of studying kinase signaling, especially for spatially resolved analyses. We demonstrated these by generating a DDR-focused MKS for monitoring the DNA damage response kinases ATR, ATM, and CHK1 and by capturing their activity over distinct cellular locations. We showed that through the use of specific targeting elements, ProKAS can resolve sub-organellar locations, such as sites of DNA synthesis within the nucleus. In these spatially resolved analyses, since the locations are pre-determined by validated targeting elements, data analysis and interpretation are straightforward. In the future, we expect that the generation of libraries targeting elements for a range of specific cellular locations will enable the systematic spatial analysis of the action of multiple kinases simultaneously. Barcoding can also be further expanded through the use of codes comprised of 4 or 5 amino acids to enable the multiplexed analysis of dozens of locations, though the design of these barcodes thus far favors the usage of small and electrostatically benign amino acid residues to minimize the possibility of altering kinase specificity and/ or aberrant peptide behavior. For example, inclusion of numerous F or W residues may dramatically alter factors like ionization, retention time, and kinase recognition due to changes in hydrophobicity or steric clashing. Even with such design principles in mind, other barcodes must be tested to ensure they do not disrupt the detectability, inducibility, or specificity of kinase sensor peptides, as demonstrated in Supplementary Fig. 9. The tandem sensor arrays may also be able to accommodate additional 10-15 amino acid sensor sequences to

monitor, in principle, dozens of kinases at once. The major factor limiting the number of sensors is likely the size of the ProKAS construct, as very large constructs would likely lead to reduced overall expression and potential structural artifacts like aggregation that may occlude sensors, prevent their phosphorylation, or even disrupt endogenous signaling networks. Removal of GFP or optimization of linker sequences in the current version of ProKAS would free up additional space to accommodate dozens of sensors. There may also be situations in which co-expressing multiple ProKAS biosensors, each with fewer (or even just one) kinase sensor peptide, would be preferable to expressing all kinase sensors in one excessively long polypeptide, which would also afford investigators more of a "mix and match" approach to ProKAS experiment design if deemed appropriate. We also demonstrated that the ProKAS system can be employed in high-throughput pipelines through semi-automated cell lysis, affinity capture, and sample preparation, allowing analyses of samples from 24-well plates on a QE-HF MS instrument. From cell treatment to quantitative readout, the experimental pipeline could often be completed in less than two days. We predict that the use of more sensitive MS instrumentation should allow the use of 96-well plates, ultimately enabling screening experiments in which the impact of entire drug collections or other conditions can be tested for their impact on kinase signaling.

Of importance, our work demonstrated the feasibility of generating sensors that distinguish the specificities of closely related kinases, such as ATR and ATM, which are known to have a similar preference for S/T-Q motifs[57]. Interestingly, the specificities conferred by the 10 amino acids surrounding the phosphorylation site of the ATR and ATM sensors we generated here were based on phosphoproteomic data and are not predicted to be highly specific according to existing in vitro preference data (the FANCD2 sensor candidate featuring a $Log_2$(Score) of 2.3 for ATR but 2.9 for ATM, for example)[20]. This inconsistency may be due to the in vitro approach relying on positional scanning peptide arrays (PSPA) where only one amino acid in the library is fixed at a time, excluding likely combinatorial effects of fixing multiple positions. This may also partially explain the low success rate of our pipeline for de novo generation of sensor sequences, as seen in the CHK1 probe development (2 positive sequences out of 10 tested). We therefore expect that sequence information embedded in the 10–15 amino acid sequence, beyond what is shown by the PSPA data, determines kinase specificity, and that such sequence determinants can be exploited in kinase sensor design. In this direction, the sensor design process may be improved by incorporating machine learning approaches trained on large datasets of experimentally validated substrates. Additional properties of sensor candidates could also be considered in the computational design, including the predicted chromatographic retention times and fragmentation patterns, which are properties for which other predictive tools have proven effective[58,59]. Regardless of the design process used to arrive at a given sensor candidate, thorough experimental validation must be performed to ensure the sensor exhibits desired quantitative behavior while also avoiding unintentional properties like low stability or degron-like behavior, though we acknowledge that such properties may in fact be useful in some biological contexts. In any case, we expect future work to expand ProKAS to other kinases through a combination of experimental and computational design processes, though design of sensors for phosphatases and tyrosine kinases will rely primarily on experimental design due to the lack of PSPA data for such enzymes. Any investigation of less-characterized kinases will require a way to probe the specificity of that kinase, whether through inhibition, knockdown, or in vitro experimentation.

The application of ProKAS to the study of ATR, ATM, CHK1, and CDKs provided insights into how these kinases respond to different genotoxic drugs. The quantitative kinetic data defined major differences in how different modes of replication stress and DNA damage promote distinct profiles of activation for each of these kinases. We foresee that systematically expanding these analyses to large panels of drugs will aid in understanding the mechanisms of drug action with greater detail. It will also be important to expand the analyses to panels of cancer cells as well as untransformed cells to define how the impact of genotoxic drugs changes depending on cell types and oncogenic state. However, in performing these expanded studies, it will be essential to ensure that the ProKAS biosensor continues to operate without any significant disruption of endogenous kinase signaling. While we have shown that the kinase sensors do not impact endogenous DDR signaling or overall cell viability in the contexts explored in this study, the possibility remains that some cell lines may react differently. Studies in such contexts must confirm that factors like biosensor size and expression level do not impact endogenous signaling before drawing firm conclusions from any data gathered. We also detected a cytosolic mode of ATM signaling induced by drugs that cause DSBs. Further work will be necessary to define the nature of this cytosolic signaling, including whether it originates in the nucleus and propagates to the cytosol, or if it originates at other organelles, such as the mitochondria. Another possibility is that DNA fragments may leak from the nucleus or the mitochondria and activate a specific pool of cytosolic ATM molecules. Interestingly, ATM is known to have a non-canonical mode of activation triggered by oxidative stress, raising the possibility that the cytosolic ATM signaling we observed could be caused by some form of oxidative stress caused by CPT and DOXR[60]. We predict that ProKAS will be useful to study the ATM response to oxidative stress with quantitative kinetics and spatial resolution.

In conclusion, ProKAS is a tool for quantitative and multiplexed analyses of kinase action with spatial resolution. The possibilities of integrating ProKAS with high-throughput analyses position this technology as a promising tool for systems biology research and drug discovery. Future developments in computational design, expanded substrate libraries, and integration with other omics technologies will further enhance the utility and impact of ProKAS in unraveling the complexities of kinase biology.

## Limitations

While ProKAS provides a flexible approach for monitoring kinase activity in living cells, it currently involves the overexpression of biosensors from plasmid libraries, which in turn requires that cells effectively uptake and express exogenous DNA. HEK293T and HCT116 cells are amenable to common transfection protocols and thus the flexibility afforded by plasmid cloning proved valuable for initial development of the ProKAS technique, but many cell lines pose challenges for achieving efficient transfection. We predict that stable expression of ProKAS biosensors would provide advantages over transient transfection, including consistency of expression and lack of transfection reagent-induced toxicity. Also, while high levels of expression achieved by common transfection may benefit sensitivity in sensor detection, drastic overexpression could increase the chances of dominant negative effects. Importantly, targeting elements must be chosen with care and rigorously validated to also avoid cytotoxic effects. Overcrowding specific subcellular loci like the plasma membrane may lead to changes in cell behavior, thereby causing aberrant signaling to be detected. ProKAS is also limited in that it monitors kinase activity in a cell population and, in contrast to FRET-based kinase probes, does not have the ability to generate data from single cells. Moreover, while the design of sensors was successful for the kinases featured in this study, the possibility remains that several kinases may require more than just the 10 flanking amino acid residues to recognize and phosphorylate a sensor sequence with specificity; for any such kinase, the ProKAS approach to sensor design would require consideration of additional variables like substrate structure, binding partners, docking sites, or prior post-translational modification of substrate sequences. Any quantitative results obtained should be

interpreted keeping in mind that sensor phosphorylation levels represent a net signaling result arising from the actions of both kinases and phosphatases (and possible sensor degradation, though not observed in this study), which can be further dissected via sensor design as demonstrated by the ATM/PPM1D kinase sensor peptide. Additionally, the possibility remains that some signaling may be picked up by the sensors as they are trafficked to the destination dictated by their targeting element, though with targeting elements like the NLS and NES used in this study, the sensors appear to dwell in their destinations for far longer than they take to get there. Optogenetic techniques exist to probe this in future studies, including light-activated NLS and other signals that could potentially provide mid-traffic signaling insight[61].

## Methods

### Plasmid construction

All pMKS plasmids constructed in this study were generated using Gibson Assembly using HiFi DNA Assembly Master Mix (NEB). PCRs were performed with primers ordered from Integrated DNA Technologies (IDT) and Q5 High-Fidelity 2X Master Mix (NEB). All gBlock synthetic gene fragments (targeting elements and MKS modules) were ordered from IDT and cloned into a plasmid containing EGFP under the control of a CMV promoter (Addgene plasmid 46957). Full plasmid sequences in .fasta format are available as Supplementary Data 9.

### Cell culture and transfection

HEK293T, U2OS, and HCT116 cells (ATCC) were cultured in DMEM (Gibco) supplemented with MEM Non-essential amino acids (Corning), Penicillin-Streptomycin (Sigma), and dialyzed Fetal Bovine Serum (Sigma). For SILAC cultures, DMEM for SILAC was used instead (Thermo Scientific), supplemented with either normal isotopes ("Light" channel) or heavy isotopes of Lysine and Arginine ("Heavy" channel). Isotopes of Lysine and Arginine used were L-Lysine-13C6,15N2 HCl and L-Arginine-13C6,15N4 HCl (Sigma). Cultured cells were passaged using 0.05% Trypsin-EDTA solution (Gibco). Reverse transfection of HEK293T cells with ProKAS plasmids was performed using 1 milligram per milliliter polyethylenimene and DMEM for SILAC to which no amino acids or serum was added ("SFS"). For a 12-well plate, 7.2 μg of plasmid DNA was mixed with 900 microliters of SFS, after which 36 microliters of 1 milligram per milliliter PEI was added (5:1 ratio of PEI:DNA) (Thermo Scientific). This mixture was allowed to incubate at room temperature for 20 minutes. Cells were detached with trypsin and counted using a Cytosmart cell counter (Corning) before being suspended in SILAC media to which the mixture of PEI, plasmid, and SFS were added prior to plating. Cells were allowed to grow for 36-48 hours before treatment. In SILAC experiments, cells grown in the Heavy channel were treated with the experimental condition unless otherwise specified. HCT116 cells were transfected with a similar procedure, substituting PEI with Mirus TransIT reagent (Mirus Bio). When using 6-, 24-, and 96-well plates, proportions of reagents were scaled with the surface area of plates and wells used. For every ProKAS experiment performed, three separate cell cultures were prepared and transfected per experimental condition in order to provide three biological replicates for every measurement made and displayed.

### Manual affinity purification

Upon completion of cell treatments, media was aspirated from the plates and 150 microliters of modified radioimmunoprecipitation assay (mRIPA) lysis buffer containing 0.2% SDS was added to the cells before plates were placed on a hot plate set to 200 °C for 10 seconds. The lysis buffer and cells were agitated before adding 450 microliters of mRIPA lysis buffer to dilute the SDS concentration below 0.1% SDS. Modified RIPA buffer was 50 millimolar Tris-HCl pH 8.0, 150 millimolar NaCl, 1% tergitol, 5 millimolar EDTA, 1X Pierce Protease Inhibitor Cocktail (Thermo Fisher CAT A32963), 1 millimolar PMSF, 5 millimolar

sodium fluoride, 2 millimolar beta-glycerophosphate, and 2 millimolar sodium pyrophosphate. For SILAC experiments, lysates for both the light and heavy SILAC channels were then pooled in 1.5 milliliter microcentrifuge tubes, whereas for label-free experiments, no pooling was performed. Lysates were then sonified using a Branson 1/8" probe tip sonifier for 5 seconds at 12% amplitude to shear genomic DNA and reduce sample viscosity. Sonified lysates were centrifuged for 5 minutes at 13,000 G+ in a 4 °C centrifuge. For each pooled lysate, 4 microliters of magnetic ALFA Selector ST affinity purification beads were equilibrated in Protein Lo-bind Eppendorf tubes (Eppendorf) containing 150 microliters of lysis buffer, after which the beads were immobilized via magnetic rack and the lysis buffer was aspirated. Centrifuged lysates were transferred to these tubes containing equilibrated beads, after which they were nutated at 4 °C for 60 minutes. After incubation with the beads, the samples were placed in magnetic racks to immobilize the beads and the supernatants were aspirated. Beads were washed twice with 500 microliters of lysis buffer and once with 1 milliliter of sterile deionized water, with each wash involving 2 minutes of nutation at room temperature. ProKAS biosensors were eluted from the beads using 80 microliters of acidic elution buffer (8 molar Urea, 0.1 molar Glycine HCl, 150 millimolar NaCl, pH 2.2) for 5 minutes at 37 °C. Elutions were transferred to new Lo-Bind tubes containing 40 microliters of 1 molar Tris pH 8.0 and 200 microliters 150 millimolar NaCl 50 millimolar Tris pH 8.0 containing 100 ng of Trypsin Gold (Promega). Tryptic digestion was allowed to proceed with nutation at 37 °C for between 12 and 18 hours before acidification with 10 microliters of 10% formic Acid (FA) and 10 microliters of 10% trifluoroacetic acid (TFA).

### Automated affinity purification

Cells were lysed the same way as for manual affinity purification, featuring the same sonification and centrifugation. Lysates were then transferred to 96-well plates with 2.2 milliliter square wells (NEST 503021). Additional plates were prepared containing ALFA beads (4 microliters of slurry into 100 microliters mRIPA lysis buffer, also containing a 96 tip comb, NEST 503311), two mRIPA lysis buffer washes (500 microliters buffer in each well), one water wash (one plate, 1000 microliters sterile water per well), and acid elution buffer (100 microliters per well). The plates were placed into a KingFisher Mag-MAX Express 96 instrument configured with BindIt 4.1 software to perform affinity purification for up to 96 samples at a time. The method file is included in the Supplementary materials. Elutions were transferred from the 96-well plate to PCR tube strips using a multi-channel pipette where they were subject to tryptic digestion with conditions identical to that of manual purification.

### MS sample preparation

Acidified tryptic digests were desalted using 20 milligrams of C18 resin extracted from 200 milligram SepPak cartridges (Waters) placed into Pierce Micro Spin columns (Thermo Scientific). Resin was conditioned with 150 microliters of 80% acetonitrile (ACN) 0.1% TFA and equilibrated with 150 microliters of 0.1% TFA before applying acidified tryptic digests. Resin was washed with 350 microliters of 0.1% acetic acid before peptides were eluted into a 1.5 milliliter microcentrifuge tube with 100 microliters of 80% ACN 0.1% acetic acid. All steps were performed using centrifugation at 800 G for 60 seconds. Elutions were dried via vacuum concentrator and resuspended in 2 microliters of LC-grade water before autosampler injection for LC-MS/MS analysis. For samples labeled with TMT, dried elutions from the described desalting protocol were instead resuspended in 8 microliters of 200 millimolar HEPES at pH 8.5 before being added to 50 microgram aliquots of TMT reagent resuspended in 2 microliters of ACN. Labeling reactions were performed for 1 hour at room temperature, after which 1 microliter of 5% hydroxylamine was added to each sample for 15 minutes to quench the labeling reaction. 150 microliters of 1% FA 5% ACN was added to

each reaction, after which all 6 reactions were pooled prior to desalting with the described desalting protocol.

## MS data acquisition

Data-dependent analyses for spectral library building were performed on a Q Exactive HF Orbitrap mass spectrometer using a 75 minute method containing a 40 minute reverse-phase gradient. This method featured a full MS scanning resolution of 60,000, an AGC target of 3e6, a maximum IT of 30 milliseconds, and a scan range of 380 to 1800 m/z. Data-dependent Top-20 MS2 parameters featured a resolution of 15,000, an AGC target of 1e5, a maximum IT of 100 milliseconds, an isolation window of 2.0 m/z, and an NCE of 28. For TMT-labeled peptides, an alternate method featuring an isolation window of 1.0 m/z and an NCE of 36 was utilized. Parallel reaction monitoring (PRM) analyses for quantitative analyses were performed on the same instrument using a 33 minute method containing a 10 minute reverse-phase gradient. This method featured an MS2 resolution of 15,000, an AGC target of 2e5, a maximum IT of 120 milliseconds, an isolation window of 0.4 m/z, and an NCE of 28. Inclusion lists featured scheduling with 1.5 minute wide retention time windows. A list of all quantification methods used for each experiment can be found in Supplementary Data 6.

## Phosphoproteomics

Pellets were subject to nuclear enrichment via resuspension in 2 milliliters of hypotonic lysis buffer and incubated on ice for 20 minutes. Nuclei were pelleted via centrifugation for 5 minutes at 1000 G. The nuclear pellets were then resuspended in 2 milliliters of mRIPA lysis buffer and lysed via sonication using a Branson Probe tip sonifier set to 15% amplitude with three 5-second pulses. Lysates were then transferred to 50 milliliter ultracentrifuge tubes for 30 minutes of centrifugation at 45,000 G kept at 4 °C. Bradford assays were performed to measure protein concentration, after which lysates for each SILAC channel were pooled for a total volume of 3 milliliters. Pooled lysates were then denatured and reduced with 1% SDS and 5 millimolar DTT at 42 °C for 15 minutes, and then alkylated with 25 millimolar iodoacetamide at room temperature for 15 minutes. Lysates were mixed with 10 milliliters of cold PPT solution (49.9% EtOH, 50% acetone, 0.1% acetic acid) to precipitate on ice for 30 minutes, after which precipitated protein was pelleted via centrifugation at 4000 G for 5 minutes. Pellets were resuspended with 1 milliliter of deionized water containing 10 microliters of Urea/Tris solution (8 molar urea, 50 millimolar Tris pH 8.0). Suspensions were transferred to clean ultracentrifuge tubes and centrifuged at 45,000 G for 5 minutes. Supernatant was aspirated before 1 milliliter of deionized water was gently added to the ultracentrifuge tube and "rolled over" the pellet three times to remove residual contaminants. This water was aspirated and the pellet itself was resuspended in 2 milliliters of Urea/Tris solution. The resuspended pellet was transferred to a 15 milliliter centrifuge tube and 6 milliliters of NaCl/Tris solution was added (150 millimolar NaCl, 50 millimolar Tris pH 8.0). This resuspension was sonified for 10 seconds at 15% amplitude to break up pieces of precipitated protein. 40 microliters of 1 milligram per milliliter TPCK-treated trypsin was added, after which the tube was parafilmed and nutated at 37 °C for 12 hours.

## Database searching

Raw MS/MS spectra were searched using the Comet search engine (part of the Trans Proteomic Pipeline 7.0.0; Seattle Proteome Center) over a composite human protein database consisting of the *Homo sapiens* proteome and the sequence of the ProKAS module[62]. Search parameters allowed for semi-tryptic peptide ends, a mass accuracy of 15 ppm for precursor ions, variable modifications for SILAC lysine and arginine (8.0142 and 10.00827 daltons, respectively), variable modification for STY phosphorylation (79.966331 daltons), and a static mass modification of 57.02146 daltons for alkylated cysteine residues[63]. TMT experiments also included static mass modifications of 229.162932 daltons on N-termini and lysine residues. Phosphorylation site localization probabilities were determined using PTMProphet, with SILAC quantification of identified phosphopeptides being performed using XPRESS (both modules part of the Trans Proteomic Pipeline; Seattle Proteome Center)[64]. Phosphoproteomic data was subject to Bowtie filtering as previously described in Faca et al.[63]. TMT quantification was performed by running Libra with the TMT 6plex channel masses specified in the condition.xml file included as Supplementary Data 11[65]. Processed phosphoproteome data are available in Supplementary Data 3 and 4. Library-building DDA data searched similarly, with search results being imported into Skyline (MacCoss Lab) for library curation.

## Skyline data analysis

PRM data files were converted to mzXML format via msConvert (Proteowizard) and imported into Skyline documents featuring parameters that are present in the Sky.zip file included in the supplement[66,67]. Transitions were checked manually for co-elution before exporting a report as a.csv containing Protein, Peptide Modified Sequence, Ratio-LightToHeavy, Normalized Area, and Replicate. Data in these reports were then processed such that SILAC ratios for each phosphopeptide were normalized against the SILAC ratios for their unphosphorylated counterparts to account for channel abundance differences. When standard deviations appeared especially low after analysis, peak areas and SILAC ratios were checked for each individual replicate via plots like that of Supplementary Fig. 13 (accompanying raw peak areas in Source Data). For label free experiments, peak areas for phosphorylated peptides were first divided by peak areas for unphosphorylated peptides in every biological replicate to provide a psuedostoichiometric quotient. The quotients for treated samples were then compared to that of untreated samples to acquire quantitative ratios. The average of the quantitative ratios for each of the three biological replicates performed for every experimental condition was then calculated, after which error bars and envelopes were applied to reflect the standard deviation of the three biological replicates.

## Fluorescence microscopy

U2OS cells (ATCC) were transfected and incubated for 24 hours before treatment with 10 micromolar 5-Ethynyl-2'-deoxyuridine (EdU; Sigma, 900584) for 15 minutes. Cells were washed once with ice-cold phosphate-buffered saline (PBS) and fixed with 4% paraformaldehyde (PFA; Sigma, 158127) for 10 minutes at room temperature. Following fixation, cells were washed twice with PBS, permeabilized with 0.5% Triton X-100 in PBS (Sigma, T9284) for 10 minutes, and subsequently washed twice with 3% bovine serum albumin (BSA; Roche Diagnostics, 03117332001) in PBS. EdU detection was performed using click chemistry, where cells were incubated with a reaction mixture containing 10 millimolar sodium-L-ascorbate (Sigma, A7631), 0.1 millimolar Atto 594 azide (Sigma, 72998), and 5 millimolar copper (II) sulfate (Sigma, C1297) for 30 minutes. The reaction solution was removed, and cells were washed three times with PBS, followed by incubation with 5% BSA for 30 minutes at room temperature. Next, cells were incubated with anti-GFP antibody (1:200; Santa Cruz, sc-9996) overnight at 4 °C, followed by three washes with PBS coverslips were mounted using Vectashield with DAPI (Vector Laboratories, H-2000) and sealed with nail polish. Imaging was performed on a Leica DMI8 microscope equipped with an HC PL APO 100x/1.40 OIL objective. Image acquisition and processing were conducted using Leica Application Suite X (LASX) software (ver. 3.8.2.27713). Colocalization analysis between ProKAS-PCNA and EdU signals was performed using Fiji/ImageJ (NIH). A randomly drawn line across the nucleus was used to generate intensity profiles via the Plot Profile function in Fiji/ImageJ. Data were then imported into GradPad Prism 9 for visualization.

## De novo design of CHK1 kinase-specific sensor

To identify optimal CHK1 sensor peptide sequences, we employed a genetic algorithm leveraging PSPA-based kinase preference scores. The algorithm aimed to maximize the predicted phosphorylation of a given 10-amino acid sequence by CHK1 while minimizing its preference for other kinases in the PSPA dataset[20]. We initialized the algorithm with a population of 50-200 randomly generated sequences and allowed it to evolve over 10-50 generations. In each generation, the fitness of each sequence was evaluated based on its PSPA scores for all 303 kinases in the dataset. The fittest sequences were then selected for reproduction through crossover and mutation operations, creating a new population for the next generation. This iterative process led to the convergence of the population towards sequences with high predicted CHK1 specificity. The algorithm was run multiple times, varying the start population size and the number of generations, to explore a wide sequence space. The resulting list of optimized peptides was further curated to ensure amino acid diversity among the final 10 selected candidates. Each sequence was flanked by AGA and R residues to ensure efficient trypsin digestion and generation of unique detectable tryptic peptides. A synthetic gBlock encoding the 10 candidate peptide sequences was obtained from IDT and cloned into a ProKAS vector containing the 53BP1 nuclear localization signal (NLS) to target the expressed biosensor to the nucleus.

For experimental validation, HEK293T cells were transfected with the ProKAS vector containing the multiplexed CHK1 sensor candidates. Following transfection, cells were treated with 1 millimolar HU for 2 hours to induce CHK1 activation. Cell lysates were collected, and the ProKAS sensors were immunoprecipitated using anti-ALFA beads. The immunoprecipitates were then trypsin digested, and the resulting peptides were analyzed by mass spectrometry. The phosphorylation levels of each candidate peptide were quantified, and the peptide demonstrating the highest inducibility and specificity upon CHK1 activation was selected as the final CHK1 sensor. A similar process was followed to acquire ATR and ATM sensors for comparison with phosphoproteome-derived sensors.

## Immunoblotting

Cells were harvested and lysed in mRIPA lysis buffer as described above. Lysates were cleared by centrifuging at 13,000 G for 5 minutes and then mixed with 3X SDS sample buffer (90 millimolar Tris-HCl pH 7.5, 3% SDS, 30% glycerol, 0.03% bromophenol blue, 60 millimolar DTT). 10% and 14% SDS-PAGE gels were run at 30 milliamps per gel before performing a wet transfer to polyvinulidene difluoride (PVDF) membranes. Membranes were incubated with primary antibodies overnight at 4 °C before being developed using BioRad Clarity ECL substrate and ChemiDoc Imaging system. The following antibodies were used: anti-KAP1 Bethyl A700-014-T, anti-pKAP1-S824 Bethyl A304146AT, anti-CHK1 Santa Cruz Biotechnology SC-8408, anti-pCHK1-S345 Cell Signaling 2341S, anti-Beta-Actin Proteintech 66009-1-Ig, anti-ALFA Rabbit IgG Fc-fusion NanoTag N1583, anti-Beta-Tubulin Proteintech 66240-1-Ig.

## MTS assay

HEK293T cells were reverse transfected as described above and seeded into a 96-well plate at a density of 10,000 cells per well in 100 microliters of medium. One column was left blank (no cells) to serve as a background control. After 24 hours, 100 microliters of camptothecin (CPT) solution was added to each well to achieve final concentrations ranging from 0 to 5000 nanomolar. Following 40 hours of CPT treatment, 100 microliters of medium was removed from each well, and 10 microliters of CellTiter 96® AQueous One Solution Cell Proliferation Assay reagent (MTS; Promega, cat. no. G3580) was added. Plates were incubated at 37 °C in a humidified atmosphere with 5% $CO_2$ for 3 hours, after which absorbance was measured at 490 nm using a BioTek Synergy H1 microplate reader

(Agilent) and BioTek Gen5 software (version 3.11, Agilent). Relative cell viability was calculated by subtracting the average absorbance of the blank wells from each reading and normalizing to the absorbance of untreated control wells. Raw plate readings are available as Supplementary Data 7.

## Cytosolic fractionation protocol

The cytosolic fraction from cell pellets was obtained by incubating tryptically detached cells in 200 microliters of hypotonic lysis buffer (HLB) on ice for 10 minutes. HLB contained 10 millimolar HEPES pH 7.9, 10 millimolar potassium chloride, 1.5 millimolar magnesium chloride, 0.34 molar sucrose, 10% glycerol, 0.1% Triton X-100, 1 millimolar dithiothreitol, 1X Pierce Protease Inhibitor Cocktail (Thermo Fisher CAT A32963), 1 millimolar PMSF, 5 millimolar sodium fluoride, 2 millimolar beta-glycerophosphate, and 2 millimolar sodium pyrophosphate. The fractionated cells were centrifuged at 1,300 G for 5 minutes at 4 °C, after which the supernatant was taken as the cytosolic fraction and diluted with 500 microliters of mRIPA lysis buffer. ALFA purification, tryptic digestion, C18 desalting, and PRM analysis was performed as previously described.

## Reporting summary

Further information on research design is available in the Nature Portfolio Reporting Summary linked to this article.

## Data availability

The mass spectrometry data generated for this study have been deposited in the PRIDE database via PXD identifier PXD065795[68]. Targeting element and MKS sequences are included in Supplementary data (Supplementary Data 1 and 2). Phosphoproteome results tables and a summary of CHK1 sensor candidate testing are also included in Supplementary Data (Supplementary Data 3, 4, and 5). Method files for execution of automated ALFA affinity purification on a KingFisher and MS acquisition are included as Supplementary Data 8. Whole plasmid sequencing results for all ProKAS plasmids are included in .fasta format as Supplementary Data 9. A Skyline document configured with all peptide and transition settings used is included as Supplementary Data 10. Source data are provided with this paper.

## Code availability

Python code and accompanying files required for generating sensor candidates in silico are are accessible via Supplementary Code 1.

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

## Acknowledgements

We thank Beatriz S. Almeida for technical support and members of the Smolka Lab for valuable discussions. This work was supported by grants from the National Institute of Health, R35GM141159 and R01HD095296 to MBS, and F31CA281247 to WJC. This manuscript is the result of funding in whole or in part by the National Institutes of Health (NIH). It is subject to the NIH Public Access Policy. Through acceptance of this federal funding, NIH has been given a right to make this manuscript publicly available in PubMed Central upon the Official Date of Publication, as defined by NIH.

## Author contributions

Conceptualization, W.J.C., M.V.A.S.N., M.B.S.; Data Curation, W.J.C., M.V.A.S.N., D.V.M., Y.R., M.W.; Formal Analysis, W.J.C., M.V.A.S.N., D.V.M., Y.R., M.W.; Funding acquisition, W.J.C, M.B.S; Investigation, W.J.C., M.V.A.S.N., D.V.M., Y.R., M.W., K.J.; Methodology, W.J.C., M.V.A.S.N., M.B.S.; Project Administration, M.B.S.; Resources, MBS; Software, M.V.A.S.N.; Supervision, M.B.S.; Validation, W.J.C., M.V.A.S.N., D.V.M., Y.R., M.W.; Visualization, WJC, M.V.A.S.N., M.W., Y.W.; Writing - original draft, W.J.C., M.V.A.S.N., M.B.S.; Writing - review and editing, W.J.C., M.B.S.

## Competing interests

The authors declare no competing interests.
