## [Transparent Peer Review file · Nature Communications]

Proteomic Sensors for Quantitative Multiplexed and Spatial Monitoring of Kinase Signaling

Corresponding Author: Professor Marcus Smolka

Version 0:

Reviewer comments:

Reviewer #1

(Remarks to the Author)

In the manuscript by Comstock et al., the authors develop overexpressed protein constructs to act as protein kinase sensors within specific locations in mammalian cells. The motivation behind this paper is the notion that understanding the temporal and perhaps spatial activity of protein kinases could further define their roles in biology and disease. The concept behind the ProKAS system is intuitive – generate substrate libraries that can be expressed in cells and therefore cover a region of the kinome. The ProKAS molecules themselves consist of 10-15mer peptides containing kinase motif peptides, fused to a GFP, localization signal and an affinity tag for retrieval. Theoretically, if a kinase is active, it will phosphorylate said motif peptide(s) in the cell of interest and that event will remain in place and correlated with the degree of activity once cells are lysed, recovered, processed and detected/analyzed by mass spectrometry. The authors demonstrate this with several examples in the DNA damage/repair pathway, including substrates that should respond to ATR, CHK1 and ATM. In these examples, they use highly validated peptide motifs – and therefore motifs/proteins that themselves would act as a kinase “sensor” – demonstrate the ability to express each, detect changes in phosphorylation under stimuli and quantify by LCMS proteomics. The examples are straightforward and support the hypothesis for the ProKAS concept, but do not result in significant new information about protein kinase activity or regulation.

Overall, this system is clever, but it is not clear that the approach is widely applicable given the information necessary to identify highly validated peptide motifs requires the existence of pre-existing annotation – as the target sequences here do. In the cases of ATR, CHK1 and ATM, the authors relied on extensive characterization in the literature from peptide arrays and target specific studies. What is confusing is that if those sites are such faithful reporters of target kinase activity (e.g., FANCD2 site in Figure 3 is such a good reporter of ATR) then why can't standard phosphoproteomics analysis of that site be used as a reporter for ATR activity alone? Why do you need an overexpressed, non-natural peptide system? What is learned that couldn't be learned here (altered subcellular localization could equally be studied by altering FANCD2 localization if that was desired)? For sites/kinase that are not so well-studied, how would this system be applied to explore them? Moreover, in all of these cases, this approach requires many biological processes to essentially be 'silent.' Each polypeptide itself could have significant biological activity (they are biologically regulated kinase motifs after all), will be subject to inherent proteolysis (in phosphorylated and non-phosphorylated states) as well as kinetic control by phosphatases. Therefore, it is challenging to conclude that one can infer kinase activity directly from the recovered levels of a motif ProKAS by MS. For example, if under two conditions one observes an increase and a decrease in phosphorylated ProKAS module, you could infer increased and decreased kinase activity, respectively. But it could also be changes in translation, proteolysis, phosphatase activity or the activity of multiple kinases that yields this change. How could this be accounted for across so many diverse polypeptide sequences? Especially unknown target kinase sequences? Those are two major conceptual limitations of the current approach that reduce the overall utility of this approach as demonstrated. For these reasons I believe this study would be more appropriate in a proteomics-focused or field-specific journal.

Other comments & questions:

- 1) Kinase and phosphatase selectivity and activity are subject to concentration dependent regulation (i.e., K_m). There is no mention of how expression level could impact these parameters in the ProKAS system and whether it could impact false positive and false negative readouts. If you use different promoters, do you see different regulation? In addition, why use transfection, where expression will be highly heterogeneous?
- 2) There is no mention of phosphatase susceptibility when considering motif selection. Or the biological consequence and

regulation of the chosen sites. Many phosphosites are degrons for example, have you tried one that is known to cause degradation to see how it behaves in this system? This seems to be a potential advantage of ProKAS modules to study downstream regulation in the absence of the native protein substrate.

3) The use of localization signals makes perfect sense, but how could one control for the modification of a nascent polypeptide on the way to or from a sub-cellular locale? The examples shown are subtly changed. Are there more specific/significant examples you can show where subcellular localization has a significant effect on phosphorylation?

4) How can you look at substrates/kinases that are not already highly characterized? This would be an advantage for the system and opportunity if possible.

Reviewer #2

(Remarks to the Author)

In this manuscript, the authors introduce an innovative approach, ProKAS, for spatially probing kinase activities by combining genetically encoded, barcoded peptide sensors with subcellular targeting and mass spectrometry. Unlike traditional subcellular proteomics, which relies on organelle fractionation or proximity labeling to infer protein localization, ProKAS enables direct, quantitative measurement of kinase activity in specific compartments. By embedding distinct localization signals and barcodes into each sensor, the platform circumvents the need for extensive biochemical purification and offers a scalable, multiplexed strategy for dissecting compartmentalized signaling responses.

While the method shows considerable promise, the work presented in this manuscript would benefit from further development to fully establish its robustness and generalizability. In particular, the demonstrated multiplexity is limited to three sensors (or six if counting different compartmental localizations), even though the approach could, in principle, be easily scaled up. Additionally, for each kinase, only a single sensor design was tested, leaving open the question of how alternative sensor designs might perform and whether they would yield consistent results. Given that this is a novel platform, addressing these scalability and robustness issues would be important to fully establish its utility.

Major Issues

1. There are many candidate peptide sequences available, yet only one was selected for building each sensor. It is unclear how sensors derived from different peptides would behave. Including comparisons under physiological conditions would be informative and would help address the robustness of the technology. For example, do sensors derived from different peptides show similar differences between signaling responses in the nucleus and cytosol (Fig. 5D)? This type of comparison is difficult to perform with fluorescent biosensors, which are much harder to design, but should be feasible with the ProKAS system.

2. While the authors report an IC_{50} of ~ 9 nM for AZD0156 using their ProKAS ATM sensor (Fig. 4G), this value is notably higher than previously reported biochemical and cellular IC_{50} values, including the reference cited (~ 0.58 nM).

Benchmarking ProKAS-derived IC_{50} values against phosphorylation of endogenous substrates under matched experimental conditions would further strengthen the platform's validation.

3. The study demonstrates two complementary approaches to ProKAS sensor design: empirical phosphoproteomic selection for ATR and ATM, and *in silico* PSPA-guided optimization for CHK1. It would further strengthen the work to compare both strategies for the same kinase (for example, testing a PSPA-designed ATR sensor alongside FANCD2 S717) to assess the consistency, sensitivity, and generalizability of the computational design pipeline relative to empirical site selection.

4. In addition, it would be informative to evaluate how the FANCD2 S717 peptide ranks using the PSPA-based scoring strategy described in Figure 3. This analysis could help unify the empirical and *in silico* approaches and provide further validation for the selection of the FANCD2 site as a preferred ATR substrate.

5. Figure 5D presents time-resolved phosphorylation profiles for ATR, ATM, and CHK1 sensors under various genotoxic conditions, but the standard deviations appear unrealistically small, with some time points showing no visible variability.

Given the expected biological and technical variability in signaling responses and mass spectrometry quantification, this raises questions about the number and nature of replicates (technical vs. biological) and the statistical handling of the data.

Clarifying these aspects and providing raw or normalized replicate values would increase confidence in the reproducibility of the reported kinetic trends.

6. In Figure 5, it is unclear whether the NLS- and NES-targeted sensor versions were cotransfected into the same cells, mixed, or separately treated. More generally, I am not sure that cloning all sensors into the same construct is necessary. This approach is labor-intensive and limits the scalability of multiplexing. An alternative would be to transfect each sensor (or different ProKAS containing a small group of sensors) separately and pool the cells before treatment. Transfection efficiency should be similar across populations and likely higher for smaller constructs. Since each sensor is internally controlled by comparing phosphorylated to unphosphorylated forms, and the assay ultimately measures bulk phosphorylation levels, separate transfections would not compromise data quality. This strategy would greatly facilitate scaling up the number of sensors and allow more flexible design of sensor combinations for different experiments.

7. As mentioned above, the claim of multiplexity would be greatly strengthened by demonstrating the ability to scale beyond three sensors.

Minor Issues

1. In Figure 2G, the data presentation is visually confusing, especially compared to Figure 2F, which clearly displays fold-changes between phosphorylated and unphosphorylated peptides using bar graphs with error bars (which are missing in Figure 2G). A similar issue is present in Figure 3G.

2. In Figure 3A, the figure suggests a brute-force search over the entire 20^9 ($\sim 5.12 \times 10^{11}$) sequence space, which would be computationally intensive. The Methods section indicated that a genetic algorithm was used for sequence optimization. It would help readers if this is made more explicit in the figure legend or main text.

Reviewer #3

(Remarks to the Author)

Reviewer #4

(Remarks to the Author)

This manuscript by Comstock et al. presents an interesting targeted proteomics approach termed ProKAS (Proteomic Kinase Activity Sensor) and demonstrates several applications of ProKAS in measuring kinase activities with spatio-temporal resolution. Overall, we find the concept novel and consider it a strong candidate for Nature Communications. However, the manuscript currently exhibits weaknesses that should be thoroughly addressed in a revision.

In general, the authors have introduced an intriguing approach or method. However, we have concerns regarding the terminology "platform," as it implies a more streamlined application process than currently described. Developing specific ProKAS sensors still requires extensive validation and optimization, thus it might be misleading to characterize the current approach as a fully developed platform. Additionally, certain experimental details need clarification, and the manuscript would significantly benefit from providing guidelines, deeper discussion, and an expanded evaluation of methodological limitations.

Major:

1. Line 177 and Figure 3E: Among the 10 candidate peptide sensors, only two phosphorylated peptides were identified upon genotoxic stress. The authors interpret this result as strong evidence supporting the importance of K/R at position -3, but this interpretation is puzzling and not fully convincing. First, why were only these two peptides detected, which also appear to represent missed cleavage events (Figure 3E)? Could this result suggest that the digestion step requires further optimization? The authors should consider using additional motif analysis or PSPdb analysis to substantiate their conclusions. Were the other eight non-phosphorylated peptides detected in genotoxic-stressed cells? The authors should improve the experimental approach or clarify the presentation in this section.

2. Figures 2F and 3F: It is unclear why unphosphorylated peptides do not exhibit an inverse trend in these assays. Given a constant introduction of ProKAS sensor peptides, increased phosphorylation should logically correspond to decreased levels of the remaining unphosphorylated form upon treatment. The authors should present the PRM data clearly and discuss whether normalization steps could explain these observations.

3. Clarification is required regarding how authors ensure consistent expression of ProKAS constructs across different cells and experiments. This point is crucial and needs explicit explanation. Additionally, authors should clarify whether variations in expression levels have been accounted for in their normalization strategies.

4. While we generally agree that orthogonal validation methods like Western blots may not be necessary for mass spectrometry-based results, the specific spatial claims made in Figure 5D would significantly benefit from validation through complementary methods, such as immunofluorescence microscopy, Western blotting, or subcellular-fractionation-based phosphoproteomics.

5. The manuscript's general discussion on considerations and guidelines for designing ProKAS sensors and conducting related experiments needs significant enhancement. The discussion could also address:

- Strategies to improve computational peptide design
- How to avoid significant disruption of the internal kinase-substrate network
- Limitations regarding membrane-localized phosphorylation sites and phosphotyrosine sites
- Practical constraints on the length of concatenated MKS sequences to avoid potential structural issues
- Stability and degradation of sensor peptides compared to endogenous substrates.

In general, we agree ProKAS represents an exciting and valuable approach, but believe a comprehensive discussion (at least mentioning these considerations) is essential to balance the message and outline future research directions.

6. The descriptions of peptide barcodes and targeting elements (together) are unclear or potentially misleading. Specifically, the rationale behind selecting AGA, GAG, or GVG as barcodes is not explained. As we understand, any amino acids can be used? Detailed sequences for NLS or NES are also lacking and should be explicitly provided.

Minor Comments:

1. Line 43: Consider specifying "without subcellular fractionation" for fairness regarding phosphoproteomics.

2. Line 161: Clarify whether the positions of lysine (K) and arginine (R) residues should be considered to ensure that the generated tryptic peptides are suitable lengths for mass spectrometry.

3. The Methods section could be strengthened by better aligning the methods described with specific experiments. For instance, SILAC was not universally applied across all experiments.

Version 1:

Reviewer comments:

Reviewer #2

(Remarks to the Author)

The authors have conducted additional experiments and analyses, and all comments have been addressed in the revised manuscript. I therefore support the publication of this manuscript.

Reviewer #3

(Remarks to the Author)

Reviewer #4

(Remarks to the Author)

The authors have provided a strong revision that addresses most of our questions and comments. We are now supportive of its publication in NC.

There are minor errors related to Figure 2 and its corresponding text that need to be fixed:

1. Figure 2D: In the blue sequence of FANCD2, there appears to be a typo: it should include a "T" not "R" before S717? Similarly, please check all the sequences in the figures and supplementary figures.
2. Figures 2E and 2F: Figure 2E, an MS2 spectrum, is cited twice in the text, whereas Figure 2F is not cited. In addition, the description for Figure 2E in the text appears to be jumbled.

Reviewer #5

(Remarks to the Author)

I was asked to review the author's response to reviewer comments. I find the response adequately addresses the reviewer's requests and has produced an improved manuscript acceptable for publication.

RESPONSE TO REVIEWERS

Our responses are indicated in blue.

We thank the reviewers for the positive and constructive feedback. We believe their perspectives have proven valuable in significantly improving the manuscript. In this new version of the manuscript, we have made major changes and additions to improve the clarity, rigor, and impact of our work, including a new main figure and 8 new supplemental figures. Key additions are:

- As requested by reviewer #2, we added an entirely new main figure (Figure 6) featuring two new sensors and three new amino acid barcodes, all combined with TMT labeling to significantly expand the multiplexing capabilities of the ProKAS technique.*
- To address requests from multiple reviewers, we included several new supplemental figures (Figs. S5 and S6, in addition to the existing S4) evaluating the effect of kinase sensor expression on cell viability, endogenous signaling, and quantitative results.*
- The revised version of the manuscript now includes several new supplemental figures and tables showing raw peak area quantification demonstrating consistent levels of unmodified sensor peptides during multiple different drug treatments as well as consistency between biological replicates. This is in response to points raised by multiple reviewers.*
- In response to reviewer #4, a new supplemental figure was added in which subcellular fractionation was used to confirm ATM sensor phosphorylation in the cytosol.*

Many other changes and additions are featured in the revised manuscript, including more detailed methods regarding biological replicates, clearer explanations of overall biosensor design, and a significantly expanded discussion touching on limitations and design/usage considerations. We believe that the changes made have strengthened the manuscript and overall development of the ProKAS technique.

Reviewer #1 (Remarks to the Author):

In the manuscript by Comstock et al., the authors develop overexpressed protein constructs to act as protein kinase sensors within specific locations in mammalian cells. The motivation behind this paper is the notion that understanding the temporal and perhaps spatial activity of protein kinases could further define their roles in biology and disease. The concept behind the ProKAS system is intuitive – generate substrate libraries that can be expressed in cells and therefore cover a region of the kinome. The ProKAS molecules themselves consist of 10-15mer peptides containing kinase motif peptides, fused to a GFP, localization signal and an affinity tag for retrieval. Theoretically, if a kinase is active, it will phosphorylate said motif peptide(s) in the cell of interest and that event will remain in place and correlated with the degree of activity once cells are lysed, recovered, processed and detected/analyzed by mass spectrometry. The authors demonstrate this with several examples in the DNA damage/repair pathway, including substrates that should respond to ATR, CHK1 and ATM. In these examples, they use highly validated peptide motifs – and therefore motifs/proteins that themselves would act as a kinase “sensor” – demonstrate the ability to express each, detect changes in phosphorylation under stimuli and quantify by LCMS proteomics. The examples are straightforward and support the hypothesis for the ProKAS concept, but do not result in significant new information about protein kinase activity or regulation.

Overall, this system is clever, but it is not clear that the approach is widely applicable given the information necessary to identify highly validated peptide motifs requires the existence of pre-existing annotation – as the target sequences here do. **In the cases of ATR, CHK1 and ATM, the authors relied on extensive characterization in the literature from peptide arrays and target specific studies. What is confusing is that if those sites are such faithful reporters of target kinase activity (e.g., FANCD2 site in Figure 3 is such a good reporter of ATR) then why can't standard phosphoproteomics analysis of that site be used as a reporter for ATR activity alone? Why do you need an overexpressed, non-natural peptide system? What is learned that couldn't be learned here (altered subcellular localization could equally be studied by altering FANCD2 localization if that was desired)? For sites/kinase that are not so well-studied, how would this system be applied to explore them? Moreover, in all of these cases, this approach requires many biological processes to essentially be 'silent.'** Each polypeptide itself could have significant biological activity (they are biologically regulated kinase motifs after all), will be subject to inherent proteolysis (in phosphorylated and non-phosphorylated states) as well as kinetic control by phosphatases.

Therefore, it is challenging to conclude that one can infer kinase activity directly from the recovered levels of a motif ProKAS by MS. For example, if under two conditions one observes an increase in phosphorylated ProKAS module and a decrease, you could infer increased and decreased kinase activity, respectively. But it could also be changes in translation, proteolysis, phosphatase activity or the activity of multiple kinases that yields this change. How could this be accounted for across so many diverse polypeptide sequences? Especially unknown target kinase sequences?

Overall, we thank the reviewer for their comments. We have conducted several new experiments (resulting in an entirely new main figure and 8 new supplemental figures) and made several text revisions to address raised concerns and answer questions. First, we have revised the text to clarify the differences in scale and practicality when comparing full-scale unbiased phosphoproteomics to ProKAS experiments. Second, we have added a new supplemental figure showcasing cell viability readouts via MTS assay, addressing concerns about how the presence of these sensor peptides impacts overall cell health and biology. Third, a new sensor has been introduced in the new main figure 6 which serves as a specific target of the phosphatase PPM1D, and much new text has been included to discuss and clarify the impact of phosphatases on the quantitative results obtained with ProKAS. Moreover, below we separately address each of the specific points raised:

What is confusing is that if those sites are such faithful reporters of target kinase activity (e.g., FANCD2 site in Figure 3 is such a good reporter of ATR) then why can't standard phosphoproteomics analysis of that site be used as a reporter for ATR activity alone?

When performing standard phosphoproteomics, no spatial information can be gleaned from the results due to the whole cell lysis performed. Additionally, phosphoproteomics experiments like the ones performed in this study require substantially more cells than ProKAS experiments, meaning the requisite cell culture, sample preparation, and mass spectrometry analysis take much longer per experimental condition rather than the time needed to process and analyze dozens of

conditions with ProKAS. We have included more text in the discussion to clarify the difference in scale of the experiments.

Why do you need an overexpressed, non-natural peptide system?

Expressing peptide sensors in the ProKAS system allows the analysis of multiple kinases in multiple locations in a single experiment, along with the ability to scale the experiments down relative to unbiased phosphoproteomics while still quantifying kinase activity. Expressing kinase sensor peptides rather than the whole protein substrate from which their sequences were derived is also expected to reduce chances of biological interference that may impact cell behavior. In the revised manuscript, we have now demonstrated that expression of our ProKAS sensors did not impact cell viability or endogenous signaling (supplementary figures S4 and S5). Figure S6 also uses promoter truncation to illustrate how dramatic changes in expression level did not have major impacts on sensor quantification, and that the expression levels obtained from separate transfected cultures are quite consistent. This study opts for overexpression of the biosensor from plasmid DNA due to the flexibility afforded in both sensor and experimental design: it is simple to edit plasmid DNA when designing sensor sequences, and expressing multiple biosensors that localize to multiple locations is possible via co-expression of multiple plasmids. However, we acknowledge in our discussion that there are many potential benefits to stable expression of the biosensor which we hope to explore in future studies.

What is learned that couldn't be learned here (altered subcellular localization could equally be studied by altering FANCD2 localization if that was desired)?

FANCD2 is a large and functional DNA repair protein, so there are numerous benefits to using only a tiny portion of its sequence to probe spatial signaling. First, changing the localization of the entire FANCD2 protein is more likely to impact cell biology due to other portions of the protein having various activities and functional interactions. Second, FANCD2 expression levels are quite low when compared to the ProKAS biosensor, requiring that we dramatically scale experiments up in order to detect S717 as a readout for ATR activation. Third, the addition of targeting elements to endogenous FANCD2 may lead to "competition" between the endogenous localization signals within the protein and our targeting elements, which could lead to unpredictable localization patterns. Fourth, by choosing only a few amino acids from the parent protein, we are then able to express it alongside other kinase sensors in our biosensor and perform highly multiplexed analyses of kinase activities.

For sites/kinase that are not so well-studied, how would this system be applied to explore them?

This is an interesting point that we are excited to explore in the future. We have included text in the discussion detailing how generation of sensors for understudied kinases would require some experimental approach to obtain specificity data, whether through in vitro expression, genetic manipulation or acute inhibition. If such approaches are available for a kinase, then generation of sensor candidates with different localizations should be possible for kinase monitoring. As we learn

more about how to predict high quality sensor sequences, we can begin to apply those principles to kinases that lack characterization of kinase substrate preference.

Moreover, in all of these cases, this approach requires many biological processes to essentially be 'silent.' Each polypeptide itself could have significant biological activity (they are biologically regulated kinase motifs after all), will be subject to inherent proteolysis (in phosphorylated and non-phosphorylated states) as well as kinetic control by phosphatases.

New supplemental figures have been included to demonstrate that kinase expression does not impact cell viability (S5), kinase sensor quantification (S6), or endogenous signaling (existing figure S4). Figures S7, S8, and S13 (in addition to the raw quantitative data in table SD8) are also now included to show that there is no downward trend in unphosphorylated sensor quantification as more phosphorylation is observed, indicating that increased phosphorylation of the kinase sensor peptides is not leading to overall biosensor proteolytic degradation. We have included a stronger acknowledgement in our discussion that any new sensors must be tested in similar ways to ensure they are indeed biologically "inert" as we expect that some future sensor candidates could indeed impact biological processes; sensors that lead to biological effects (e.g. dominant negative effects on cellular responses) should be avoided.

Other comments & questions:

1) Kinase and phosphatase selectivity and activity are subject to concentration dependent regulation (i.e., K_m). There is no mention of how expression level could impact these parameters in the ProKAS system and whether it could impact false positive and false negative readouts. If you use different promoters, do you see different regulation? In addition, why use transfection, where expression will be highly heterogeneous?

We agree that ProKAS expression levels warrant further investigation and have thus performed numerous experiments to assess whether changing expression impacted sensor quantification or behavior. A new supplemental figure S6 is added in which we compared ProKAS readouts between cells expressing the biosensor downstream of the full CMV promoter versus a truncated CMV promoter known to drastically reduce expression. After confirming lower expression via Western Blot, we saw no major changes in sensor phosphorylation when performing mass spectrometry analysis. The blots in this figure, in addition to the raw peak area data added to the supplement, also serve to demonstrate the consistency of expression achieved using the transient transfection protocols chosen for this study. Text has also been added to the discussion detailing the advantages and disadvantages of using transient transfection in this manuscript (flexibility versus transfection reagent toxicity), while also touching on ways in which stable expression would prove useful in future studies (lack of transfection reagent toxicity, use of transfection-resistant cell lines).

2) There is no mention of phosphatase susceptibility when considering motif selection. Or the biological consequence and regulation of the chosen sites. Many phosphosites are degrons for example, have you tried one that is known to cause degradation to see how it behaves in this system? This seems to be a potential advantage of ProKAS modules to study downstream regulation in the absence of the native protein substrate.

We agree that phosphatase and protease susceptibility are valid concerns, and have taken many steps to investigate both possibilities.

- Regarding phosphatases, a sensor sequence that is both a target of the kinase ATM and the phosphatase PPM1D is utilized in the new main figure 6, showing how phosphatase activity can indeed vary from sensor to sensor based on the sequence utilized. We have recently developed this new PPM1D/ATM sensor as part of work focused on this phosphatase and have cited the preprint that was recently deposited on BioRxiv. Accompanying discussion text has also been included to clarify how the quantification obtained from ProKAS is a net signaling result, essentially an equilibrium between active kinases and active phosphatases.*
- Regarding potential degradation, new supplemental figures S7, S8, and S13 have been included showing raw peak areas for sensor peptides in multiple kinetics experiments. These figures demonstrate how the unmodified version of each kinase sensor peptide does not decrease in abundance as higher phosphorylation levels are detected, strongly suggesting that phosphorylation of the currently utilized sensor peptides does not lead to degradation of the ProKAS biosensor to a significant degree. We also find the notion of possible degron-like sensors fascinating, and have added further text to our discussion section regarding the possible utility of future sensor peptides that do in fact act as degrons.*

3) The use of localization signals makes perfect sense, but how could one control for the modification of a nascent polypeptide on the way to or from a sub-cellular locale? The examples shown are subtly changed. Are there more specific/significant examples you can show where subcellular localization has a significant effect on phosphorylation?

We thank the reviewer for bringing this consideration to our attention and have added text to the discussion regarding the possibility of sensor phosphorylation during the brief time between translation and final localization. We have also elaborated on ways in which signaling changes due to changes in traffic could be probed, namely through the use of optogenetic targeting elements. That being said, we believe that the differential localizations showcased in the study present useful examples of significant changes in phosphorylation patterns. For example, in figure 5D, there are very robust signaling responses to HU and GEM in the nucleus whereas there is zero induction in the cytosol, presenting a significant difference in signaling between those two compartments. The same is true when directing sensors to replication forks, where no induced ATM activity is detected due to this localization. We hope that future studies reveal more targeting elements that result in significant changes in sensor phosphorylation, like mitochondrial or membrane localization signals.

4) How can you look at substrates/kinases that are not already highly characterized? This would be an advantage for the system and opportunity if possible.

We agree that this is an important point worth discussing further than was initially included in the manuscript, and have thus added discussion of how to conduct such studies. In essence, any kinase could be studied with ProKAS as long as there are approaches to assess its substrate specificity, whether that be through in vitro kinase activity, chemical inhibition or genetic manipulation. With that information in hand, sensor candidates can then be generated, tested, and validated via the previously utilized mutations or inhibitors. Information regarding some limiting factors is included in the discussion, including how some kinases require docking sites or other cofactors for substrate recognition; for kinases where this is the case, docking sites or other domains would likely need to be engineered alongside the kinase sensor peptide itself.

Reviewer #2 (Remarks to the Author):

In this manuscript, the authors introduce an innovative approach, ProKAS, for spatially probing kinase activities by combining genetically encoded, barcoded peptide sensors with subcellular targeting and mass spectrometry. Unlike traditional subcellular proteomics, which relies on organelle fractionation or proximity labeling to infer protein localization, ProKAS enables direct, quantitative measurement of kinase activity in specific compartments. By embedding distinct localization signals and barcodes into each sensor, the platform circumvents the need for extensive biochemical purification and offers a scalable, multiplexed strategy for dissecting compartmentalized signaling responses.

While the method shows considerable promise, the work presented in this manuscript would benefit from further development to fully establish its robustness and generalizability. In particular, the demonstrated multiplexity is limited to three sensors (or six if counting different compartmental localizations), even though the approach could, in principle, be easily scaled up.

Additionally, for each kinase, only a single sensor design was tested, leaving open the question of how alternative sensor designs might perform and whether they would yield consistent results. Given that this is a novel platform, addressing these scalability and robustness issues would be important to fully establish its utility.

We thank the reviewer for their enthusiasm regarding the potential of the ProKAS technique, and we share the sentiment that further development and expansion would greatly strengthen the technique overall. As such, we have included a new main figure 6 that expands on the technique in three ways: the inclusion of 2 new sensors, curation of 3 new codes, and use of TMT 6plex for expanding multiplexing capabilities. Additional experiments were also performed to showcase how other obtained sensor candidates behaved, as well as the inclusion of raw peak areas for kinetic experiments to improve data transparency and shed light on replicate consistency.

Major Issues

1. There are many candidate peptide sequences available, yet only one was selected for building each sensor. It is unclear how sensors derived from different peptides would behave. Including comparisons under physiological conditions would be informative and would help address the robustness of the technology. For example, do sensors derived from different peptides show similar differences between signaling responses in the nucleus and cytosol (Fig. 5D)? This type of comparison is difficult to perform with fluorescent biosensors, which are much harder to design, but should be feasible with the ProKAS system.

We agree that showing the results from multiple sensors for the same kinase would prove insightful, and have thus included another ATM sensor in the new main figure 6 which was derived from the protein RAP80 (Serine 101). Interestingly while this sensor becomes phosphorylated to the same degree as our 53BP1-derived ATM sensor, it is a specific target for dephosphorylation by the phosphatase PPM1D. We believe this aptly demonstrates how different candidate peptides can behave similarly and differently in various regards. We also utilized the in silico sensor design approach for ATM, yielding multiple viable ATM sensor peptides, some with roughly similar

inducibility and specificity when compared to the 53BP1-derived sensor (Figure S1). In the future, it will be interesting to test if/how different localizations may differentially impact phosphorylation of different sensor sequences targeted by the same kinase, which could also reveal differential phosphatase action at distinct locations.

2. While the authors report an IC_{50} of ~9 nM for AZD0156 using their ProKAS ATM sensor (Fig. 4G), this value is notably higher than previously reported biochemical and cellular IC_{50} values, including the reference cited (~0.58 nM). Benchmarking ProKAS-derived IC_{50} values against phosphorylation of endogenous substrates under matched experimental conditions would further strengthen the platform's validation.

We thank the reviewer for their observation, and we have corrected our statement in the text regarding the cited IC_{50} to reflect the difference in values obtained. In our revision we consider that the cited IC_{50} is from a different cell line and how in vivo differences may underlie the higher value obtained from our experiment. However, we also note that the relative selectivity of the inhibitor for ATM over ATR (~1000 fold) shown in our study does align well with the selectivity observed in the cited paper.

3. The study demonstrates two complementary approaches to ProKAS sensor design: empirical phosphoproteomic selection for ATR and ATM, and in silico PSPA-guided optimization for CHK1. It would further strengthen the work to compare both strategies for the same kinase (for example, testing a PSPA-designed ATR sensor alongside FANCD2 S717) to assess the consistency, sensitivity, and generalizability of the computational design pipeline relative to empirical site selection.

We agree that the PSPA-guided sensor design approach deserves more direct comparison with the phosphoproteome-selected sensors, and thus have added a new supplemental figure (Fig. S1) in which we generated and tested 10 ATM sensor candidates using the in silico approach. Interestingly, we find that many of these candidates are inducible and specific, though the level of inducibility varies somewhat. We believe this shows the viability of the in silico approach to sensor design, and we thank the reviewer for this suggestion.

4. In addition, it would be informative to evaluate how the FANCD2 S717 peptide ranks using the PSPA-based scoring strategy described in Figure 3. This analysis could help unify the empirical and in silico approaches and provide further validation for the selection of the FANCD2 site as a preferred ATR substrate.

We agree that PSPA-based scores for the phosphoproteome-derived sensors would prove insightful, and have thus included in the discussion PSPA-based scores for the FANCD2 sensor candidate using the PhosphoSitePlus kinase prediction tool. Accompanying this, we have added significantly to the discussion regarding the PSPA approach in general, noting possible variables it does not account for that may result in scores like that of the FANCD2 sensor candidate.

5. Figure 5D presents time-resolved phosphorylation profiles for ATR, ATM, and CHK1 sensors under various genotoxic conditions, but the standard deviations appear unrealistically small, with some time points showing no visible variability. Given the expected biological and technical variability in signaling responses and mass spectrometry quantification, this raises questions about the number and nature of replicates (technical vs. biological) and the statistical handling of the

data. Clarifying these aspects and providing raw or normalized replicate values would increase confidence in the reproducibility of the reported kinetic trends.

We have included supplemental figures showing the raw peak areas acquired for the CPT and HU kinetics datasets in figure 4 (figures S7 and S8) as well as peak areas and SILAC ratios for the nuclear ATM sensor during HU treatment from figure 5, selected as an example from amongst the many sensors and drug treatments in panel D (figure S13). Raw peak area values for these figures are now also included in supplementary table SD8. Presentation of these raw peak areas and SILAC ratios add transparency to the manuscript and clarify the consistency of measurements obtained. Additionally, the methods section has been expanded to provide further clarity regarding how every ProKAS measurement was acquired from three separate biological replicates.

6. In Figure 5, it is unclear whether the NLS- and NES-targeted sensor versions were cotransfected into the same cells, mixed, or separately treated. More generally, I am not sure that cloning all sensors into the same construct is necessary. This approach is labor-intensive and limits the scalability of multiplexing. An alternative would be to transfect each sensor (or different ProKAS containing a small group of sensors) separately and pool the cells before treatment. Transfection efficiency should be similar across populations and likely higher for smaller constructs. Since each sensor is internally controlled by comparing phosphorylated to unphosphorylated forms, and the assay ultimately measures bulk phosphorylation levels, separate transfections would not compromise data quality. This strategy would greatly facilitate scaling up the number of sensors and allow more flexible design of sensor combinations for different experiments.

We agree that the sensor expression scheme in figure 5 was unclear and have revised the text and figure legend to clarify that the NLS- and NES-containing biosensors were co-transfected into the same cell cultures. We also agree with the reviewer regarding the potential advantages of performing separate transfections followed by pooling, and hope that our application of separate transfections followed by pooling in the new main figure 6 reflects some of those advantages. Text has been added to the discussion to describe some of the other advantages of discrete sensor expression including the experimental design flexibility afforded by such an approach.

7. As mentioned above, the claim of multiplexity would be greatly strengthened by demonstrating the ability to scale beyond three sensors.

We strongly agree with the reviewer regarding scalability and have thus expanded the multiplexing capabilities of the ProKAS technique in three different dimensions per this suggestion: 3 new codes, 2 new sensors, and multiplexed TMT labeling/quantification. This data is now presented in an entirely new main figure (figure 6). Verification of the new codes has been included in the supplement (figure S9), alongside validation of a new pan-CDK sensor peptide (figure S12) and inclusion of a new ATM/PPM1D sensor peptide we recently developed as part of another work from our lab that was recently deposited as a preprint. TMT 6plex labeling has allowed us to increase experimental multiplexing, as demonstrated by quantifying 5 sensors in 36 conditions from a single sample. In future work, our goal is to expand the multiplexing capabilities by adding over a dozen of new barcodes to allow analysis of hundreds of samples in a single MS run.

Minor Issues

1. In Figure 2G, the data presentation is visually confusing, especially compared to Figure 2F, which clearly displays fold-changes between phosphorylated and unphosphorylated peptides using bar graphs with error bars (which are missing in Figure 2G). A similar issue is present in Figure 3G.

We agree that the data presentation could be more intuitive, and have revised the mentioned panels to better reflect the quantification of sensor inducibility and specificity. These revisions have been consistently applied to Figures 2, 3, and S2.

2. In Figure 3A, the figure suggests a brute-force search over the entire 20^9 ($\sim 5.12 \times 10^{11}$) sequence space, which would be computationally intensive. The Methods section indicated that a genetic algorithm was used for sequence optimization. It would help readers if this is made more explicit in the figure legend or main text.

We have amended the figure legend to better reflect the genetic algorithm approach to prevent the misconception that billions of sensor sequences were individually scored.

Reviewer #3 (Remarks to the Author):

Reviewer #4 (Remarks to the Author):

This manuscript by Comstock et al. presents an interesting targeted proteomics approach termed ProKAS (Proteomic Kinase Activity Sensor) and demonstrates several applications of ProKAS in measuring kinase activities with spatio-temporal resolution. Overall, we find the concept novel and consider it a strong candidate for Nature Communications. However, the manuscript currently exhibits weaknesses that should be thoroughly addressed in a revision.

In general, the authors have introduced an intriguing approach or method. However, we have concerns regarding the terminology "platform," as it implies a more streamlined application process than currently described. Developing specific ProKAS sensors still requires extensive validation and optimization, thus it might be misleading to characterize the current approach as a fully developed platform. Additionally, certain experimental details need clarification, and the manuscript would significantly benefit from providing guidelines, deeper discussion, and an expanded evaluation of methodological limitations.

We thank the reviewers for their interest in the ProKAS technique, and agree that the manuscript could benefit from clarification and deeper discussion. Numerous points in the text have been revised and expanded thanks to the input provided. Additionally, multiple new experiments have been performed to address concerns, such as effects of expression levels, and to validate biological results, including the cytosolic ATM signaling observed in figure 5. When taken with the new sensor peptides and multiplexing capabilities, we hope that the revised manuscript provides a clearer explanation of the ProKAS technique in its now more developed state.

Major:

1. Line 177 and Figure 3E: Among the 10 candidate peptide sensors, only two phosphorylated peptides were identified upon genotoxic stress. The authors interpret this result as strong evidence

supporting the importance of K/R at position -3, but this interpretation is puzzling and not fully convincing. First, why were only these two peptides detected, which also appear to represent missed cleavage events (Figure 3E)? Could this result suggest that the digestion step requires further optimization? The authors should consider using additional motif analysis or PSPdb analysis to substantiate their conclusions. Were the other eight non-phosphorylated peptides detected in genotoxic-stressed cells? The authors should improve the experimental approach or clarify the presentation in this section.

We have clarified the accompanying text to speculate on why the overall success rate in obtaining a de novo CHK1 sensor was relatively low. Essentially, the in vitro approach relies on positional scanning peptide arrays (PSPA) where only one amino acid in the library is fixed at a time, excluding likely combinatorial effects of fixing multiple positions that might be vital for kinase recognition. Additionally, the current algorithm makes no attempt to optimize for ionization efficiency of sensor candidates given the current lack of such predictive capabilities; it follows that many generated sensor sequences may ionize poorly and remain undetected by our instrumentation. More detail on how the algorithm could be improved has been provided, including mention of both retention time and ionization efficiency prediction (if such tools ever become available). The results for the 10 CHK1 sensor candidates are also included in supplemental table SD5, which indicates that 9 candidates were detected in their unphosphorylated forms while only 2 were detected in their phosphorylated forms.

2. Figures 2F and 3F: It is unclear why unphosphorylated peptides do not exhibit an inverse trend in these assays. Given a constant introduction of ProKAS sensor peptides, increased phosphorylation should logically correspond to decreased levels of the remaining unphosphorylated form upon treatment. The authors should present the PRM data clearly and discuss whether normalization steps could explain these observations.

We agree that this was worth clarifying further and have included two supplemental figures (S7 and S8) showing raw peak areas (without normalization) for CPT and HU kinetics experiments to showcase how unphosphorylated peak areas do not deplete as sensors themselves increase in phosphorylation. Peak area values are also included in new supplementary table SD8. We have also included further text to draw attention to the small percentage of the overall sensor population that becomes phosphorylated (low stoichiometry), causing negligible impacts on the remaining unmodified population of the sensors.

3. Clarification is required regarding how authors ensure consistent expression of ProKAS constructs across different cells and experiments. This point is crucial and needs explicit explanation. Additionally, authors should clarify whether variations in expression levels have been accounted for in their normalization strategies.

We agree that expression levels are an important variable to account for and have thus added a supplemental figure (S6) that examines the impact of reduced expression on the quantitative results obtained from the ProKAS technique. In this figure we also show western blot analysis of three separately transfected cell populations for each promoter we utilized, which show consistent biosensor abundance across the biological replicates. We hope that this bolsters confidence in the consistent level of expression of the ProKAS biosensors, though we strongly agree with the reviewer that alternate methods of expression (like stable cell lines) are worth exploring and would feature many benefits mentioned in the discussion ranging from reduced transfection reagent-induced toxicity to more consistent expression levels to longer treatment duration monitoring. More detail has also been provided in the methods section describing how induced phosphorylation is

calculated via normalizing phosphorylated peptides by their unphosphorylated counterparts, thereby accounting for any variability in expression or purification efficiency.

4. While we generally agree that orthogonal validation methods like Western blots may not be necessary for mass spectrometry-based results, the specific spatial claims made in Figure 5D would significantly benefit from validation through complementary methods, such as immunofluorescence microscopy, Western blotting, or subcellular-fractionation-based phosphoproteomics.

We agree that the cytosolic signaling seen in Figure 5 is worth probing further and have thus added a supplemental figure (S11) in which we performed cellular fractionation after expressing the cytosolic ProKAS biosensor and retrieving the cytosolic fraction for sensor quantification. Sensor phosphorylation levels obtained from the cytosolic fraction closely mimic that seen in figure 5, which we believe provides strong evidence that the ATM sensor phosphorylation is indeed taking place in the cytosol under those conditions.

5. The manuscript's general discussion on considerations and guidelines for designing ProKAS sensors and conducting related experiments needs significant enhancement. The discussion could also address:

- Strategies to improve computational peptide design
- How to avoid significant disruption of the internal kinase-substrate network
- Limitations regarding membrane-localized phosphorylation sites and phosphotyrosine sites
- Practical constraints on the length of concatenated MKS sequences to avoid potential structural issues
- Stability and degradation of sensor peptides compared to endogenous substrates.

In general, we agree ProKAS represents an exciting and valuable approach, but believe a comprehensive discussion (at least mentioning these considerations) is essential to balance the message and outline future research directions.

We agree and have significantly augmented the discussion section of the manuscript to elaborate on all of these points:

- *Incorporating consideration for peptide ionization, retention time, and other deep-learning resources for computational peptide design.*
- *Emphasis on empirical testing of sensor candidates in similar ways to what is shown in this study (cell viability, blotting for endogenous signaling markers, etc.) to ensure internal signaling is not disrupted*
- *Acknowledgement that PSPA data could not be currently used to make tyrosine kinase substrates, but that experiment-derived tyrosine kinase sensors should still be possible following the rationale utilized to obtain ATR and ATM sensors in this study. Potential overcrowding of membranes is also noted for any future interest in membrane-targeted biosensors.*
- *Noting that excessively long MKS modules may lead to structural artifacts like aggregation that could occlude kinase substrates, emphasizing that expression of multiple shorter MKS modules may be a way to circumvent these issues if they ever arise.*
- *Noting that some sensor candidates may be unstable or act as degrons, further increasing the importance of empirical validation of any new sensors. Sensors in this study showed no drop in unmodified sensor abundance as phosphorylation increased (figures S7, S8, S13, and table SD8), though it remains a possibility for future sensor design endeavors.*

6. The descriptions of peptide barcodes and targeting elements (together) are unclear or potentially misleading. Specifically, the rationale behind selecting AGA, GAG, or GVG as barcodes is not explained. As we understand, any amino acids can be used? Detailed sequences for NLS or NES are also lacking and should be explicitly provided.

We agree that more rationale regarding the amino acid barcoding is warranted. More detail is provided upon the introduction of amino acid barcoding along with more information about the testing and considerations involved in future barcode selection in the discussion. We specifically point out that current barcode designs favor smaller and electrostatically benign residues, avoiding highly charged residues or bulky hydrophobic residues like F or W which could change sensor behavior biologically or analytically. Sequences for the NLS, NES, and PCNA targeting elements are also included in supplemental table SD1.

Minor Comments:

1. Line 43: Consider specifying "without subcellular fractionation" for fairness regarding phosphoproteomics.

We agree with this and have added to the specified text.

2. Line 161: Clarify whether the positions of lysine (K) and arginine (R) residues should be considered to ensure that the generated tryptic peptides are suitable lengths for mass spectrometry.

We have added text to the text accompanying main figure 1 detailing this important consideration, noting that having proximal K or R residues on both sides of a kinase target would result in tryptic peptides too small to reliably detect and quantify.

3. The Methods section could be strengthened by better aligning the methods described with specific experiments. For instance, SILAC was not universally applied across all experiments.

We have added to our methods section to clarify how experiments were designed and executed, especially with respect to their quantitative methodologies and replicate schema. Additionally, we have included another supplemental data table (SD6) that lists every ProKAS experiment included in this study and the quantitative methodology used, either SILAC, LFQ, or TMT.